# HiVid: LLM-Guided Video Saliency For Content-Aware VOD And Live Streaming

**Jiahui Chen**[1], **Bo Peng**[1], **Lianchen Jia**[1], **Zeyu Zhang**[2], **Tianchi Huang**[1], **Lifeng Sun**[1,3,*]

[1]Tsinghua University, [2] The Australian National University
[3]Key Laboratory of Pervasive Computing, Ministry of Education, [*] Corresponding Author
chenjiah22@mails.tsinghua.edu.cn, sunlf@tsinghua.edu.cn

## ABSTRACT

Content-aware streaming requires dynamic, chunk-level importance weights to optimize subjective quality of experience (QoE). However, direct human annotation is prohibitively expensive while vision-saliency models generalize poorly. We introduce HiVid, the first framework to leverage Large Language Models (LLMs) as a scalable human proxy to generate high-fidelity weights for both Video-on-Demand (VOD) and live streaming. We address 3 non-trivial challenges: (1) To extend LLMs' limited modality and circumvent token limits, we propose a perception module to assess frames in a local context window, autoregressively building a coherent understanding of the video. (2) For VOD with rating inconsistency across local windows, we propose a ranking module to perform global re-ranking with a novel LLM-guided merge-sort algorithm. (3) For live streaming which requires low-latency, online inference without future knowledge, we propose a prediction module to predict future weights with a multi-modal time series model, which comprises a content-aware attention and adaptive horizon to accommodate asynchronous LLM inference. Extensive experiments show HiVid improves weight prediction accuracy by up to 11.5% for VOD and 26% for live streaming over SOTA baselines. Real-world user study validates HiVid boosts streaming QoE correlation by 14.7%.

## 1 INTRODUCTION

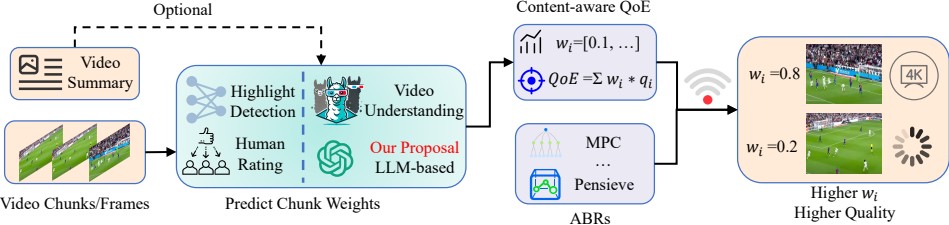

Figure 1: Overview of content-aware streaming. The estimated chunk weights $w_i$ are incorporated into QoE and optimized by ABRs. Higher weights would render better viewing experience.

Content-aware video streaming improves quality of experience (QoE) by allocating higher bitrates to more important video chunks guided by user-perceived priority weights Zhang et al. (2021). As shown in Figure 1, with available video chunks and optional text description, we can estimate the saliency score and incorporate it into existing QoE model. Following past work on highlight detection Moon et al. (2023); Xiao et al. (2024), here we denote the saliency as the overall content importance score for each video chunk. We distinguish it from visually salient regions within a frame in classic video saliency prediction tasks.

The adaptive bitrate (ABR) algorithms Chen et al. (2024a) then optimize the bitrates with preference priority such that higher weights incur higher quality and less rebuffering, thus rendering better subjective experience. However, such human-centric and content-dependent saliency task brings new challenges to existing paradigms.

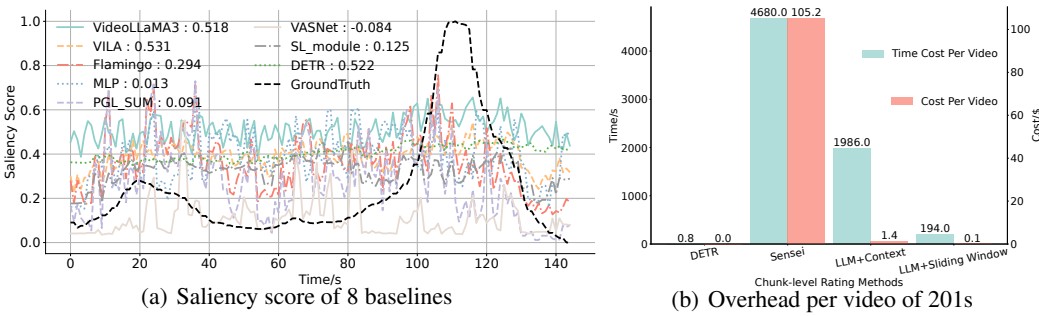

(a) Saliency score of 8 baselines

(b) Overhead per video of 201s

Figure 2: Inaccurate saliency of previous work and significant overhead of human ratings.

**Challenge 1**: *Why LLM and Its Constraints.* The most intuitive solution is computer vision (CV) based highlight detection like DETR Moon et al. (2023), which learns to identify per-chunk temporal saliency scores from training videos. However, these models are too small to capture the complex semantic content and generalize across diverse video categories. Alternatively, large video understanding models like VideoLLaMA3 Zhang et al. (2025) excel in objective question answering and captioning tasks, but they often yield invalid and inaccurate responses when it comes to zero-shot subjective rating. We present a case study in Fig. 2 (a) (refer to experimental setup). We can find that neither paradigm can fit the ground truth with high PLCC correlation (legend value). On the contrary, SENSEI Zhang et al. (2021) conducts offline crowdsourcing ratings with human involvement, which is accurate but expensive and time consuming (78 minutes and 100$ per video). Therefore it's impractical for large-scale deployment, especially for live streaming, as shown in Fig. 2(b).

To enable both accuracy and efficiency, we can harness LLMs for zero-shot subjective reasoning as human proxy. However, video modality is unavailable for most SOTA LLMs like GPT-4o, which motivates us to assess only anchor frames from each chunk. Moreover, the limited input tokens (e.g., 128k) prohibit memorizing all historical context when dealing with long videos (LLM+Context in Fig. 2(b)). Therefore, we can break down the frames via local sliding window to enable fine-grained rating and global summarization with minimal overhead (LLM+Window in Fig. 2(b)).

**Challenge 2**: *Rating Discrepancies in video-on-demand (VOD).* Due to the lack of global context across frames in Challenge 1, the LLM rating distribution may vary significantly across different local windows. We present an example in Fig. 3 with sliding window length $m = 10$. During frames 79-80, the scoring scene receives the most attention (the highest ground truth) but the rating only reaches 65-70, while the less intense celebration during frames 108-109 is rated 75-85. This is because the results from one window only manifest local importance without global reference. To enable consistency, an appropriate re-ranking across chunks can eliminate context bias and rating discrepancy.

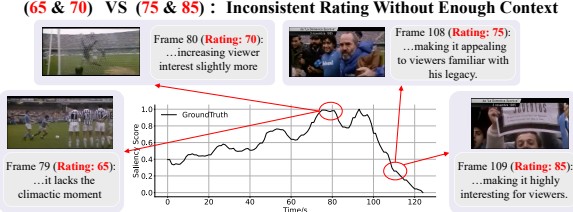

Figure 3: Inconsistent rating distribution.

**Challenge 3**: *Uncertainty in Live Streaming.* Different from VOD, live streaming requires real time decision without future chunks' knowledge. To this end, we can only predict the future weights based on historical ratings via forecasting models. However, the LLM inference latency is variable and dependent on the input tokens (see Table 2). Therefore, a robust prediction must adjust the future horizon to cover the time interval gap for chunks that are not yet rated. Only then can ABRs optimize the future chunks' weighted QoE to decide the optimal bitrate. In addition, the inherent multi-modal time series also calls for a new content-aware forecasting model to further boost accuracy.

In response, we propose HiVid, the first systematic framework that harnesses the power of LLMs as judge for content-aware streaming with 3 tailored modules. To address challenge 1, we propose a perception module to derive overall video description and chunk-level saliency scores. We leverage LLMs to assess sampled anchor frames from each chunk via a local sliding window. The response

comprises frame group ratings with periodical video summary as a compact historical context for subsequent windows. In this way, HiVid is adaptive to arbitrary video length without token limits.

To address challenge 2, we propose a ranking module on top of the previous perception. With the global video summary and group ratings, we propose to re-rank the groups with a novel variant of merge sort algorithm, which encompasses an LLM-guided comparison capable of sorting multiple frames. In this way, we obtain a globally consistent saliency map without distribution discrepancy, while the overall summarization from perception module also guides the LLM reasoning.

To address challenge 3, we propose a prediction module in parallel with perception module. Upon each response of previous group rating, we leverage a novel multi-modal time series forecasting model to predict the future chunk weights that are yet to arrive. We align frames and periodical text summary with CLIP Radford et al. (2021), and then we propose a content-aware attention to capture the impact of multi-modal video statistics on time series evolution. To further meet the strict latency, we dynamically adjust the prediction dimension asynchronously depending on LLM and forecasting latency. In this way, we achieve real-time streaming by pre-generating the future weights.

We conduct extensive experiments on 3 well-known highlight detection datasets. Regarding VOD, HiVid surpasses 8 SOTA highlight detection and video understanding models by 11.5%, 6% and 14.7% in terms of correlation, mean average precision (mAP) and mean opinion score (MOS) accuracy. Regarding live streaming, HiVid also outperforms 9 SOTA forecasting by 26% while guaranteeing real-time latency. We summarize our contributions as follows:

• We present HiVid, the first coherent LLM-guided pipeline for content-aware VOD and live streaming. We identify 3 key challenges: (1) Constrained LLM modality and context length; (2) Rating distribution discrepancy in VOD; (3) Unavailable future chunks and strict latency requirement in live streaming.

• We address the issues with 3 modules: (1) Perception module that assesses sampled frames via context windows to iteratively generate video summary and saliency scores; (2) Ranking module that leverages LLM-guided merge sort algorithm to re-rank all the frames with global video summary. (3) Prediction module that leverages a multi-modal time series model to predict future weights, compounded by a novel content attention mechanism and adaptive forecasting dimension.

• HiVid achieves the SOTA across 17 baselines in extensive experiments on public datasets. Real world user study in streaming QoE also validates the effectiveness.

## 2 RELATED WORK

### 2.1 CONTENT-AWARE STREAMING

Traditional video streaming leverages ABRs like heuristic MPC Yin et al. (2015) to decide bitrates of chunks to maximize objective QoE metrics Duanmu et al. (2019), i.e. higher visual quality, lower rebuffering, etc. Content-aware streaming Zhang et al. (2021) improves upon additive chunk-level QoE Mao et al. (2017) by incorporating the subjective content preferences as:

$$QoE = \sum_{i}^{N} w_i * q_i \tag{1}$$

where $w_i$ and $q_i$ denote the chunk weight and objective metrics above. To derive $w_i$, SENSEI Zhang et al. (2021) leverages crowdsourcing ratings on videos with different low-quality chunks and then infers the optimal weights. However, such human rating process amounts to expensive cost with significant delay, which is not scalable for VOD and live streaming.

As an alternative, highlight detection Xu et al. (2021) like DETR Moon et al. (2023) tends to predict the chunk saliency score from video features by training neural networks like transformers. Video summarization Apostolidis et al. (2021) like VASNet Fajtl et al. (2019) achieves a similar goal by inferring chunk importance to the whole video. However, these small models exhibit poor semantic understanding and generalization ability, especially for unseen videos. Recently, large video models like VILA Lin et al. (2024) have enhanced the performance of various understanding tasks. However, they suffer from hallucination and often yield invalid and inaccurate responses when dealing with subjective but quantitative rating task.

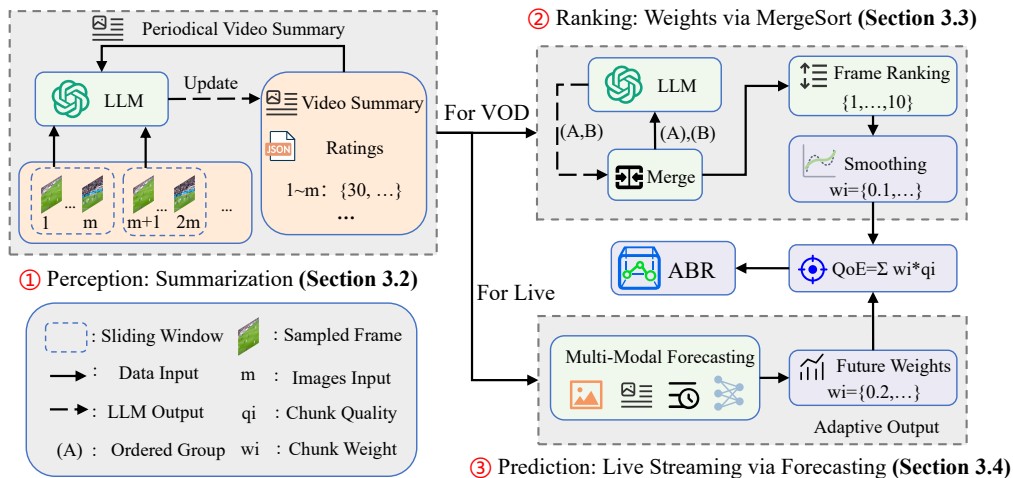

Figure 4: Overview of HiVid. The perception module generates a video summary with group ratings. The ranking module yields a ranking list via a variant merge sort algorithm for VOD streaming. The prediction module predicts future weights via adaptive forecasting for live streaming. The final weights $w_i$ are incorporated into the QoE model.

## 2.2 LLMs FOR SUBJECTIVE REASONING

On the contrary, LLMs Achiam et al. (2023) have exhibited better semantic reasoning compared with video understanding models. LLMs have been adapted as an agent Ge et al. (2023) to perform various understanding Jin et al. (2024) and scheduling tasks Lai et al. (2023), while several studies Park et al. (2023); Hussain et al. (2024); Liu et al. (2025) also demonstrate the correlation between LLMs and human behavior regarding subjective perception assessment. However, it has not been explored how LLMs empower video-level highlights rating because video modality is not directly supported, and the limited context prohibits an entire video input, presenting a significant gap.

## 3 PROPOSED METHOD

### 3.1 OVERVIEW OF HIVID

Built upon previous insights, we present our novel framework HiVid which comprises 3 modules in Fig. 4. The perception module quickly iterates through the video and generates a summary with group ratings via sliding window. To adapt to VOD and eliminate rating discrepancies, the ranking module leverages an LLM-guided merge sort algorithm to rank all the frames, with guidance from previous summary. The final smoothing further refines the oscillating ratings. To adapt to live streaming with latency constraints, the prediction module utilizes adaptive multi-modal forecasting to predict future weights in an asynchronous manner. Together the 3 modules enable efficient and effective content-aware video streaming.

### 3.2 (BASIC) PERCEPTION MODULE

In response to **Challenge 1**, we propose to leverage LLM to understand and rate the video chunks via sliding window. To align with image modality in LLMs, we directly sample the anchor frame as the first frame of each chunk to reduce redundancy and computation overhead, while other sampling like the last frame also suffice (see Appendix F). Unless specified, we estimate the chunk weight with the sampled frame rating. For a video of $D$ chunks and window of $m$ length, we upload the $m$ frames along with periodical summary to the LLM. The prompt instructs (see Appendix J) the LLM to rate the $m$ images based on existing context and then update the summary:

$$R_{(k-1)m+1}^{km}, S_{km} = LLM(F_{(k-1)m+1}^{km}, S_{(k-1)m}), k \in [1, \left\lceil \frac{D}{m} \right\rceil]$$

(2)

where $R_i^j, F_i^j, S_i$ denote rating and frame group from $i$ to $j$ and periodical summary before $i$ frames, $j = min(j, D)$. The initial summary $S_0$ is the basic title and background of each video. In this way, we iteratively derive the overall summarization and all the frame ratings, with only $\lceil \frac{D}{m} \rceil$ LLM calls.

### 3.3 (VOD) RANKING MODULE

In response to **Challenge 2**, we propose to re-rank the grouped chunks to eliminate context discrepancies. To this end, we leverage a variant merge sort algorithm but with LLMs as the comparison function, which is capable of sorting $m$ frames in $O(m)$ time.

**Merging Two Groups.** Built upon perception module, to merge two sorted group frames, $A = SF_1^{n_1}$ and $B = SF_1^{n_2}$, we pick $\frac{m}{2}$ frames from each group to form a new $m$ list for sorting. We then extract the first $\frac{m}{2}$ sorted frames and put the rest back to the original group, which can be formulated as:

$$(SF_{k_1}^{k\frac{m}{2}}, SF_{k\frac{m}{2}}^{k_m}) = LLM(SF_1^{\frac{m}{2}}, SF_1^{\frac{m}{2}}, S_D) \tag{3}$$

where $SF_i^j$ denotes sorted frames from $i$ to $j$, $S_D$ is the overall summary from Equ. 2. When either group is exhausted, we directly append the remaining sorted frames to the final list. By repeating Equ. 3 until groups $A$ and $B$ are both exhausted, we derive the final sorted $n_1 + n_2$ frames.

**Sorting All Groups.** For a video of $D$ chunks and $\lceil \frac{D}{m} \rceil$ groups of no more than $m$ frames, we first obtain the $SF$ from sorting $R$ in perception module. Then we follow typical binary recursion algorithm to iteratively merge groups to obtain sorted $D$ frames, which represent overall content preferences with global context from both frames and text summary. To evaluate the worst merging overhead, we derive the following formula:

$$T(k) = T(\lfloor \frac{k}{2} \rfloor) + T(\lceil \frac{k}{2} \rceil) + 2k - 1, k = \lceil \frac{D}{m} \rceil \tag{4}$$

where $T(k)$ is the number of LLM calls for sorting $k$ groups and $T(1) = 1$. After obtaining the two sorted halves of $D$ frames, we need to merge them into the final list. While the worst scenario is when neither half is exhausted faster than the other during Equ. 3. Therefore each LLM sorting extracts $\frac{m}{2}$, rendering $\lceil \frac{D}{\frac{m}{2}} \rceil = 2 \lceil \frac{D}{m} \rceil = 2k$ calls, except that for the last time, we can directly sort the remaining $\leq m$ frames without putting the second half back.

The $T(k)$ complexity of Equ. 4 is $O(k \log k)$, rendering total complexity of ranking module $O(k \log k) + O(k)$, where $O(k)$ is the overhead of sliding window from perception module.

**Gaussian Smoothing.** With the final ranking $SF_1^D$, we normalize the index to $[0, 1]$ as chunk weights $w_i$. To better fit the smooth ground truth distribution, e.g., in Fig. 3, we further apply Gaussian smoothing to alleviate the oscillation as $w_i = GS(s, \sigma, w_i)$, where kernel size $s = D$ and $\sigma$ is the standard deviation. We present the final algorithm in Appendix B.

### 3.4 (LIVE) PREDICTION MODULE

In response to **Challenge 3**, we propose to leverage time series forecasting to predict future weights in parallel with perception module. We illustrate the scenario in Fig. 5. Upon each frame upload, the LLM response may arrive later after a token-related interval. Therefore, to predict future $N$ weights from previously rated $m$ chunks, the output dimension should cover the time gap for previous $n - m$ chunks without ratings and $N$ future chunks. Only then can ABRs optimize Equ. 1 to decide the bitrate of chunk $n + 1$. In this way, we eliminate the significant delay by asynchronous prediction and achieve real-time streaming.

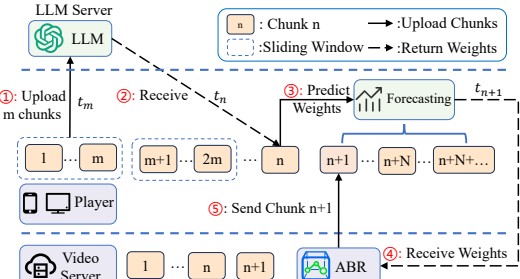

Figure 5: We predict future weights upon LLM response. The future horizon is latency-adaptive.

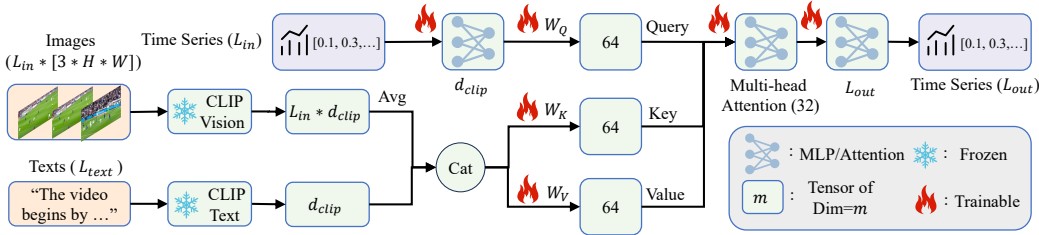

Figure 6: Multi-modal forecasting with content attention.

**Forecasting Model.** Different from traditional forecasting, we have as input not only the series data $x_w \in R^{L_{in}}$, but also historical frames $x_i \in R^{L_{in} \times 3 \times H \times W}$ and video summary $x_t \in R^{L_{text}}$. To incorporate 3 modalities, we leverage a well-known CLIP Radford et al. (2021) to align the image and text features. To capture the complex interdependent relationships, we further propose a novel content-aware attention mechanism. We project the time series features as Query (Q), and the concatenated image and text content features are projected as Key (K) and Value (V):

$$Attn(F(x_w), F(x_{cat}), F(x_{cat})) = softmax(\frac{Q_w K_{cat}^T}{\sqrt{d}}) \cdot V_{cat} \qquad (5)$$

where $F(x_w)$ denotes time series features and $F(x_{cat}) = Cat(CLIP_v(x_i), CLIP_t(x_t))$. In this way, we motivate the model to learn an attention pattern that specifically answers: *Given the historical video content, what context is most relevant if the time series weights evolve as such?* The detailed network architecture is in Fig. 6. Each previous frame is coupled with a rating as time series with length $L_{in}$, while the general video summary has constant length $L_{text}$. We leverage a frozen CLIP to derive the latent features and then average the image to align the dimension. Then we project the 3 modality features into Q, K, V in Equ. 5. The multi-head attention is followed by a linear layer and finally predicts the future weights with length $L_{out}$.

To enhance the prediction performance, typical mean squared error (MSE) loss does not suffice, because the weights distribution represents relative preference. Therefore we propose a novel correlation loss to guide the model as follows:

$$loss = MSE(x, x_{gt}) + \lambda * (1 - \frac{\sum(x - \mu_x)(x_{gt} - \mu_{x_{gt}})}{\sigma_x \sigma_{x_{gt}}}) \qquad (6)$$

where $x$ and $x_{gt}$ are the predicted and ground truth weights, $\mu_x$ and $\sigma_x$ are the mean and standard deviation.

---

**Algorithm 1:** Forecasting with adaptive output

**Input:** constant parameters $d, m, N$, current chunk number $i$, global variable future weights $W$
**Output:** Future weights $w_{i+1}^{i+N}$

1  **if** $i\%m == 0$ **then**  submit $m$ frames to LLM                           ▷ Equ. 2 ;
2  **if** *LLM response is updated* **then**
3      determine $L_{out}(d, m, N)$ by Equ. 7;
4      submit $m$ time series to Forecasting($L_{in} = m, L_{out}$)
5  **end**
6  **if** *forecasting results $w$ is updated* **then**  update $w$ into $W$ ;
7  **if** $w_{i+1}^{i+N}$ *in* $W$ **then  return** $w_{i+1}^{i+N}$                    ▷ weighted QoE in Equ.1 ;
8  **else  return** $[1] * N$                                    ▷ original QoE without $w_i$ in Equ. 1 ;

---

**Adaptive Prediction.** The core idea is to predict longer future weights that include the model inference time, as shown in Fig. 5. Therefore, the prediction dimension is adaptive to the LLM and forecasting latency. Assume each chunk is of duration $d$, we submit $m$ frames (as chunks) to LLM at $t = t_m$ in Equ. 2 but receive at time $t = t_n$, we have LLM interval $\Delta t = t_n - t_m$ and forecasting latency $\delta$, rendering elapsed chunks without rating $\lceil \frac{\Delta t + \delta}{d} \rceil$.

Table 1: Saliency accuracy of 2 method diagrams. Blue and Red denote the best and worst.

| Dataset | Metrics | Large Model-based | | | | Vision Saliency-based | | | | |
|---|---|---|---|---|---|---|---|---|---|---|
| | | HiVid | VideoLLaMA3 | VILA | Flamingo | MLP | PGL-SUM | VASNet | SL-module | DETR |
| Youtube-8M | PLCC↑ | 0.66 | 0.54 | 0.52 | 0.41 | 0.59 | 0.52 | 0.55 | 0.59 | 0.57 |
| | SRCC↑ | 0.67 | 0.55 | 0.54 | 0.41 | 0.60 | 0.54 | 0.56 | 0.60 | 0.58 |
| | mAP50↑ | 0.86 | 0.77 | 0.73 | 0.56 | 0.81 | 0.80 | 0.80 | 0.81 | 0.81 |
| | mAP15↑ | 0.53 | 0.45 | 0.44 | 0.33 | 0.49 | 0.46 | 0.46 | 0.49 | 0.45 |
| TVSum | PLCC↑ | 0.50 | 0.41 | 0.37 | 0.32 | 0.44 | 0.39 | 0.45 | 0.43 | 0.42 |
| | SRCC↑ | 0.52 | 0.41 | 0.37 | 0.30 | 0.43 | 0.40 | 0.45 | 0.43 | 0.44 |
| | mAP50↑ | 0.67 | 0.52 | 0.53 | 0.47 | 0.62 | 0.59 | 0.66 | 0.57 | 0.63 |
| | mAP15↑ | 0.40 | 0.29 | 0.31 | 0.25 | 0.38 | 0.37 | 0.34 | 0.33 | 0.33 |
| SumMe | PLCC↑ | 0.47 | 0.35 | 0.35 | 0.31 | 0.37 | 0.33 | 0.37 | 0.39 | 0.38 |
| | SRCC↑ | 0.47 | 0.35 | 0.36 | 0.30 | 0.37 | 0.34 | 0.37 | 0.39 | 0.39 |
| | mAP50↑ | 0.62 | 0.49 | 0.55 | 0.39 | 0.52 | 0.53 | 0.57 | 0.53 | 0.61 |
| | mAP15↑ | 0.37 | 0.24 | 0.33 | 0.23 | 0.31 | 0.35 | 0.33 | 0.30 | 0.32 |

Moreover, since LLMs are called every $m$ frames at $t = t_{km}$, the response also arrives periodically rather than at per frame frequency. Hence we need to secure the future weights for those without LLM call or response, i.e. chunk number $m$. Finally, ABR algorithm typically requires $N$ future chunk weights to optimize the QoE model, which incurs the final prediction dimension as follows:

$$L_{out} = \left\lceil \frac{\Delta t + \delta}{d} \right\rceil + m + N \tag{7}$$

**Live Streaming Pipeline.** The detailed process is in Algorithm 1. Since $\Delta t + \delta$ may vary dynamically, we first train several models with randomized different $L_{out}$. During inference, for the initial chunks $\left\lceil \frac{\Delta t + \delta}{d} \right\rceil + m$ without LLM response, we pad the chunk weights with default 1 (line 8). For current chunk $i$, we upload to the LLM if a group of window length $m$ is complete (line 1). This rating process is executed asynchronously from current video playback. Then we perform rating forecasting when the latest LLM response is available, which equals time interval of $\Delta t$. Given this LLM latency and estimated prediction time $\delta$, we can derive the required adaptive output by Equ. 7. Then we pick a trained model with minimum output dimension satisfying $L_{out}$ to ensure the highest accuracy (line 3). This forecasting process is also executed locally and asynchronously (line 4). Upon new future weights $w$ (from last forecasting submission), we cache the result in a global weight pool $W$ for future reference (line 6). Finally we check the latest future $N$ weights for content-aware QoE model (line 7) if available.

## 4 EXPERIMENT

**Datasets and Metrics.** We conduct experiments on Mr.Hisum from Youtube-8M Sul et al. (2023), TVSum Song et al. (2015) and SumMe Gygli et al. (2014) which includes 1953, 50, 25 videos respectively. We sample 7:1.5:1.5 for training, validation, and testing, respectively. For saliency scores, we leverage correlation-based Pearson's linear correlation coefficient (PLCC) and Spearman's rank correlation coefficient (SRCC), and we also include highlight detection metrics mAP50 and mAP15 for comprehensive comparison. For forecasting, we leverage typical mean absolute error (MAE), root mean square error (RMSE) and also PLCC and SRCC.

**Parameter Setting.** The default LLM used for HiVid is GPT-4o unless specified. Video chunks $D$ depends on the test video length, window length $m = 10$ unless specified, chunk duration $d = 1s$, Gaussian smoothing kernel size $s = D$, $\sigma = 5$, forecasting loss $\lambda = 1$, $L_{in} = m$, pretrained models with $L_{out} = \{1, 2, 3\} * L_{in}$, ABRs' decision horizon $N = 5$. We also fix the ABR as RobustMPC Yin et al. (2015) since the QoE can be dynamically adjusted by $w_i$ during each optimization. The QoE model is the same as Pensieve Mao et al. (2017) unless specified. The network trace dataset is FCC Commission (2016) and 3G/HSDPA Riiser et al. (2013) for later user study.

**8 Saliency Baselines.** We select 2 highlight detection methods SL-module Xu et al. (2021) and DETR Moon et al. (2023), 2 video summarization based PGL-SUM Apostolidis et al. (2021) and VASNet Fajtl et al. (2019) and an MLP based network. We modify the loss function to MSE to learn the exact saliency score, and we also concatenate the same Gaussian smoothing after each model for fairness. We also include 3 SOTA video understanding models, VideoLLaMA3 Zhang et al. (2025),

VILA Lin et al. (2024) and Flamingo Alayrac et al. (2022). We leverage sliding window like HiVid due to invalid response on entire video rating.

**9 Time Series Forecasting Baselines.** For uni-modal baselines, we compare HiVid-U (built on only MLPs without image and text modalities) with 6 SOTA methods, iTransformer Liu et al. (2023), TimeMixer Wang et al. (2024), TimesNet Wu et al. (2022), Crossformer Zhang & Yan (2023), PatchTST Nie et al. (2022) and FiLM Zhou et al. (2022). We also include two efficient architectures RNN Sherstinsky (2020) and LSTM Zhao et al. (2017). For multi-modal baselines, there are no methods that incorporate image modality. Therefore we compare HiVid-M (with 3 modalities) with LLM-based method, where we input all the series data, historical frames and text summary with a prompt instruction for forecasting.

## 4.1 VOD: SALIENCY SCORE EVALUATION

**Saliency Score Accuracy.** To demonstrate our perception and ranking modules, we first evaluate saliency score and present the results in Table 1. We can find that HiVid outperforms with 11.5% and 6% improvement on average PLCC and mAP50 compared with the second SL-module respectively, thanks to our video summary and robust ranking. The latest model DETR Moon et al. (2023) ranks only the middle, which demonstrates that even the SOTA

Table 2: Overhead comparison of different $m$. 2 (mini) denotes GPT-4o-mini.

| Metrics | $m$ | 2(mini) | 2 | 4 | 6 | 8 | 10 |
|---|---|---|---|---|---|---|---|
| Per API | Token Score↓ | 15 | 278 | 292 | 314 | 321 | 335 |
| | Average latency/s↓ | 3.01 | 3.14 | 4.2 | 6.4 | 8.14 | 9.83 |
| | Latency Std/s↓ | 1.46 | 1.52 | 1.8 | 0.65 | 0.54 | 0.83 |
| Per Video | Perception Calls↓ | 100 | 100 | 50 | 34 | 25 | 20 |
| | Ranking Calls↓ | 1358 | 1358 | 581 | 350 | 242 | 182 |
| | Total Cost/$↓ | 0.44 | 8.12 | 3.68 | 2.41 | 1.71 | 1.35 |
| | Total Time Cost/h↓ | 1.21 | 1.26 | 0.73 | 0.67 | 0.60 | 0.54 |

saliency method cannot fully capture video semantic content due to model scaling. In addition, video models like VILA also exhibit lower accuracy due to inferior reasoning compared with large-scale LLMs. For a more illustrative case study, we present a saliency distribution in Appendix C.

**Overhead Analysis.** We present time and monetary costs for different window lengths $m$ in Table 2. Per API call, higher $m$ means more input tokens and hence higher score and higher latency. However, higher $m$ also performs better which renders much fewer API calls as validated by Equ. 2 and Equ. 4. Therefore, the total cost per video of 201s is generally lower.

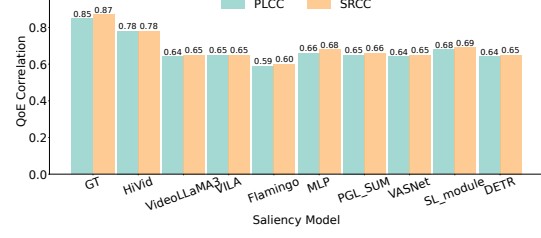

Figure 7: MOS correlation↑ of saliency models.

**User Study.** To demonstrate real world streaming performance, we evaluate the QoE when combining saliency score from 10 different baselines including ground truth. We leverage RobustMPC ABR to optimize a dynamic QoE model from Mao et al. (2017) with the saliency weights. We sample 10 category-varying test videos from Youtube-8M encoded at {300, 750, 1200, 1850, 2850, 4300} kbps. Note that we extract only 10s clips around the highest score for viewers. We run ABRs with 4 random network traces from FCC Commission (2016) and 3G/HSDPA Riiser et al. (2013). For each viewer, we have 320 10-seconds clips.

We recruit 10 volunteers to evaluate the above clips and rate each from 1 to 100. We randomly shuffle all the clips with the same video to ensure fair rating without prejudice. Finally, we compute the correlation between weighted QoE model and averaged MOS. The results are in Fig. 7. We can find that HiVid outperforms with 0.1-0.19 higher PLCC than SL-module and Flamingo, which vali-

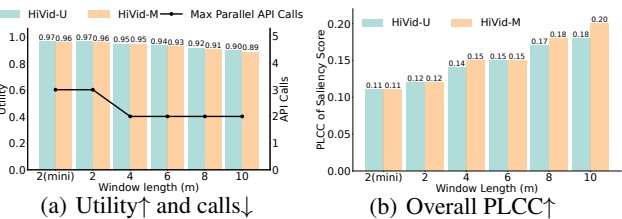

(a) Utility↑ and calls↓      (b) Overall PLCC↑

Figure 8: Forecasting performance in live streaming.

dates the high PLCC in Table 1. In summary, HiVid achieves the SOTA in both weight correlation and user experience.

Table 3: Time series forecasting w/o LLM output. The results are averaged on 3 datasets among $L_{in} = \{8, 10\}$ and for each $L_{in}$, $L_{out} = \{1, 2, 3\} * L_{in}$. For RNN and LSTM, $L_{out} = \{1\} * L_{in}$.

| Models | | | | | | Uni-Modal | | | | | Multi-Modal | |
|---|---|---|---|---|---|---|---|---|---|---|---|---|
| Metrics | | iTransformer | TimeMixer | TimesNet | Crossformer | PatchTST | FiLM | RNN | LSTM | HiVid-U | LLM | HiVid-M |
| w/ PLCC loss | MAE↓ | 0.08 | 0.08 | 0.08 | 0.09 | 0.09 | 0.09 | 0.18 | 0.18 | 0.08 | 0.13 | 0.08 |
| | RMSE↓ | 0.13 | 0.13 | 0.14 | 0.12 | 0.14 | 0.14 | 0.22 | 0.22 | 0.12 | 0.27 | 0.12 |
| | PLCC↑ | 0.15 | 0.14 | 0.15 | 0.23 | 0.15 | 0.15 | 0.09 | 0.09 | 0.24 | 0.17 | 0.29 |
| | SRCC↑ | 0.11 | 0.13 | 0.13 | 0.22 | 0.12 | 0.11 | 0.08 | 0.10 | 0.22 | 0.18 | 0.27 |
| w/o PLCC loss | MAE↓ | 0.08 | 0.08 | 0.09 | 0.09 | 0.08 | 0.09 | 0.19 | 0.18 | 0.08 | 0.13 | 0.08 |
| | RMSE↓ | 0.14 | 0.14 | 0.14 | 0.12 | 0.14 | 0.14 | 0.22 | 0.22 | 0.14 | 0.27 | 0.13 |
| | PLCC↑ | 0.09 | 0.08 | 0.09 | 0.14 | 0.09 | 0.09 | 0.05 | 0.06 | 0.16 | 0.17 | 0.21 |
| | SRCC↑ | 0.08 | 0.08 | 0.08 | 0.13 | 0.09 | 0.08 | 0.04 | 0.06 | 0.16 | 0.18 | 0.20 |
| Time/ms↓ | | 22 | 26 | 48 | 16 | 24 | 22 | 4 | 5 | 3 | 8134 | 1350 |

Table 4: Saliency accuracy of HiVid w/ open-source multi-modal LLMs.

| Metric | HiVid w/ GPT-4o | Llama-3.2-11B-Vision-Instruct | Qwen3-VL-8B-Instruct | InternVL3-14B | gemma-3-12b-it |
|---|---|---|---|---|---|
| PLCC↑ | 0.66 | 0.58 | 0.60 | 0.64 | 0.61 |
| SRCC↑ | 0.67 | 0.60 | 0.61 | 0.64 | 0.62 |
| mAP50↑ | 0.86 | 0.80 | 0.81 | 0.84 | 0.82 |
| mAP15↑ | 0.53 | 0.45 | 0.50 | 0.52 | 0.50 |

## 4.2 LIVE: FORECASTING EVALUATION

**Forecasting Metrics.** We present the time series performance in Table 3. HiVid-M outperforms all the SOTA baselines with the highest PLCC=0.29 and also the lowest MAE=0.08, thanks to our novel content-aware attention. Even in uni-modality scenario, HiVid-U also achieves the best performance with PLCC=0.24 and MAE=0.08. This demonstrates that more complex models do not necessarily lead to better performance without tailored design. In addition, the improved accuracy when combined with correlation loss in Equ. 6 has also validated our novel design.

LLM-based method performs worse because such task would require sufficient training data, rather than subjective reasoning. As for the time overhead, HiVid-U is the fastest due to simple MLP concatenation, HiVid-M may exhibit more cost but it can be circumvented by the asynchronous pipeline in Equ. 7.

**Streaming Metrics.** To demonstrate HiVid's application, we present the forecasting utility (ratio of chunks with available future weights) and overall correlation (forecasting with LLM rating as input) in Fig. 8. We can find that for higher $m$, the utility decreases due to longer initial LLM response interval in Equ. 7, while parallel calls also decrease which minimizes the risk of response blocking, i.e. early calls arriving later.

As for the overall PLCC/SRCC, they are bottlenecked by the forecasting accuracy, even with accurate historical weights. Therefore the performance is better with higher $m$, but with the upper bound from forecasting PLCC=0.29 (for HiVid-M).

For end-to-end latency in real ABR streaming, we present the time overhead in Appendix D. Overall HiVid imposes near-zero latency on original ABR, thanks to our asynchronous LLM and forecasting inference. In general, HiVid also outperforms all the baselines in forecasting accuracy and latency.

## 4.3 ABLATION STUDY

To demonstrate the generalization of HiVid, we also apply the ranking module on open-source multi-modal LLMs (mllm) Dubey et al. (2024); Chen et al. (2024b); Yang et al. (2025); Team et al. (2025) in Table 4. It is as expected that GPT-4o still outperforms with up to 13.7% PLCC improvement. However, local mllm can guarantee consistency

Table 5: Ablation of HiVid modules in VOD.

| Model | Performance | | | | Per Video | |
|---|---|---|---|---|---|---|
| | PLCC↑ | SRCC↑ | mAP50↑ | mAP15↑ | Cost/$↓ | Time Cost/h↓ |
| HiVid | 0.660 | 0.674 | 0.860 | 0.526 | 1.35 | 0.54 |
| HiVid w/ m=2 (mini) | 0.645 | 0.651 | 0.848 | 0.511 | 0.44 | 1.21 |
| Gemini-2-flash | 0.604 | 0.592 | 0.812 | 0.503 | 0.03 | 0.63 |
| Grok2 | 0.616 | 0.613 | 0.824 | 0.506 | 1.69 | 0.69 |
| Claude-3-haiku | 0.53 | 0.55 | 0.807 | 0.477 | 0.41 | 0.83 |
| HiVid w/o Perception | 0.632 | 0.653 | 0.852 | 0.520 | 1.22 | 0.49 |
| HiVid w/o Ranking | 0.611 | 0.617 | 0.820 | 0.498 | 0.13 | 0.054 |
| HiVid w/o GS | 0.619 | 0.621 | 0.835 | 0.514 | 1.35 | 0.54 |

across runs and can be further fine-tuned for specific tasks, though it requires significant local computation.

To demonstrate the effectiveness of each module, we conduct ablation study in Table 5. Note that all LLM backbones adopt window length $m = 10$ for fair comparison. We can find that HiVid (with GPT-4o, $m = 10$) outperforms with the best performance and lowest time overhead, while Gemini achieves the lowest cost, exhibiting different advantages.

By removing the perception module, HiVid cannot capture the global text summary which hinders some improvement. However, without the ranking module, there are significant rating discrepancies as shown in Fig. 3, which leads to much lower performance. Finally, the Gaussian smoothing also refines the coarse saliency score distribution to some extent.

To demonstrate our adaptive prediction, we leverage constant $L_{out}$ and append the rest with

Table 6: Overall PLCC↑ (w/ LLM rating as input) for adaptive and constant $L_{out}$.

| $m$ PLCC↑ | 2(mini) | 2 | 4 | 6 | 8 | 10 |
|---|---|---|---|---|---|---|
| HiVid-M | 0.11 | 0.12 | 0.15 | 0.15 | 0.18 | 0.20 |
| $L_{out} = m$ | 0.05 | 0.05 | 0.07 | 0.08 | 0.08 | 0.10 |
| $L_{out} = 2m$ | 0.08 | 0.09 | 0.10 | 0.12 | 0.15 | 0.17 |
| $L_{out} = 3m$ | 0.09 | 0.09 | 0.12 | 0.15 | 0.17 | 0.18 |

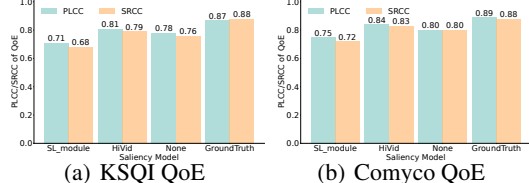

(a) KSQI QoE      (b) Comyco QoE

Figure 9: MOS correlation↑ of other QoE models.

1 when necessary as required in Equ. 7. The results in Table 6 show that neither baseline can reach our accuracy. Because dummy future weights directly degrade the correlation, while longer $L_{out}$ also means inferior model performance, given that $L_{in}$ is constant for each $m$.

To demonstrate the generalization of HiVid, we conduct additional user study with different QoE models in Fig. 9, i.e. KSQI Duanmu et al. (2019) and Comyco Huang et al. (2019). HiVid still outperforms with 0.1 and 0.09 improvement on PLCC, which stems from our content-

Table 7: LLM robustness for ambiguous videos.

| Category | PLCC | | SRCC | | mAP50 | | mAP15 | |
|---|---|---|---|---|---|---|---|---|
| | Mean ↑ | Std ↓ | Mean ↑ | Std ↓ | Mean ↑ | Std ↓ | Mean ↑ | Std ↓ |
| Politics | 0.73 | 0.05 | 0.74 | 0.05 | 0.92 | 0.03 | 0.63 | 0.04 |
| People | 0.69 | 0.03 | 0.69 | 0.04 | 0.88 | 0.03 | 0.60 | 0.02 |
| Education | 0.63 | 0.04 | 0.62 | 0.06 | 0.83 | 0.02 | 0.55 | 0.02 |

only saliency assessment. We also present more comparisons with different ABRs and parameters in Appendix E and F.

To validate the robustness against hallucination, we average the rating of 3 LLMs (GPT-4o, Gemini and Claude) on ambiguous videos. The results in Table 7 show that even sensitive categories yield stable accuracy with low deviation, thanks to our robust ranking. In general, all the modules in HiVid contribute to the overall accuracy and efficiency.

To demonstrate the large-scale application of HiVid in live streaming, we present detailed cost and PLCC of an example video of 2 hours in Table 8. Since HiVid applies sliding window for live streaming without future chunks, the cost increases linearly with video length, i.e. LLM rating and forecasting once per $m$ video

Table 8: Performance of longer live streaming.

| Window Length | Metric | 2 min | 30 min | 2 hour | 6 hour | 24 hour |
|---|---|---|---|---|---|---|
| m=10 | Cost/$ ↓ | 0.08 | 1.20 | 4.81 | 14.43 | 57.74 |
| | PLCC ↑ | 0.21 | 0.26 | 0.24 | / | / |
| m=2 (mini) | Cost$ ↓ | 0.02 | 0.27 | 1.09 | 3.26 | 13.03 |
| | PLCC ↑ | 0.13 | 0.16 | 0.15 | / | / |

chunks. For longer window lengths $m = 10$, the performance is better but with higher cost (57.74\$ per 24 hours), yielding a controllable tradeoff. However, we argue that we only need to process the source video in real-world one-to-many streaming, therefore the cost is actually negligible.

## 5 CONCLUSION

We introduced HiVid, the first systematic framework to leverage LLMs for content-aware streaming. We identify the critical trade-off between the inaccuracy of vision-based models and the prohibitive cost of human annotation. We addressed 3 core challenges: (1) a perception module that updates summary and ratings via a sliding window to extend modality and context limitations; (2) an LLM-guided ranking module that ensures globally consistent saliency scores for VOD; (3) a prediction module with content-aware model based on adaptive dimension to meet the strict real-time live streaming. Extensive experiments on public datasets and user study demonstrate our effectiveness.

## 6 ACKNOWLEDGEMENTS

This work was supported by Key Laboratory of Data Intelligence, Beijing.

## 7 ETHICS STATEMENT

This paper does not raise any ethical issues regarding human subject or dataset usage.

## 8 REPRODUCIBILITY STATEMENT

We have provided a code example of our basic ranking module in the Supplementary Material in OpenReview. It includes how to combine merge sort with LLMs and how to compute the total API calls for overhead analysis. The attached json file is an example of how to store the response of each sliding window. In this way, we can cache the previous results and resume the ranking in case of API disconnection. While the periodical video summary can also be updated by querying the last json results and uploading as LLM input.

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

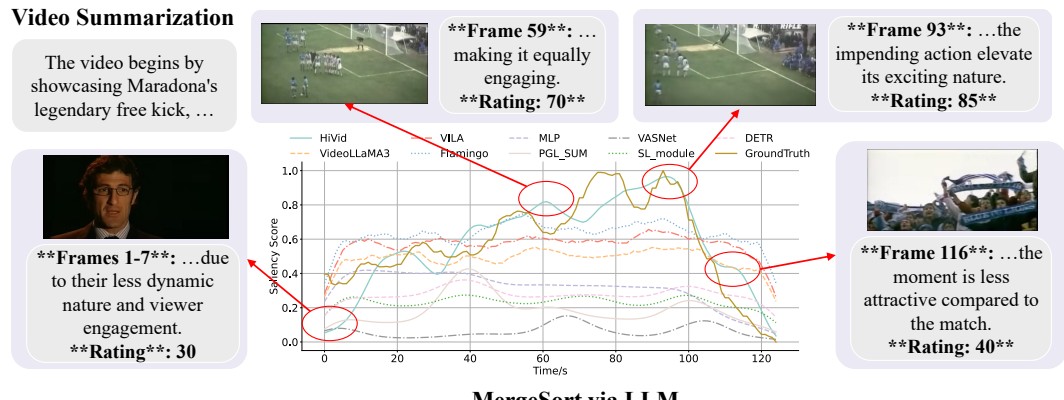

Figure 10: Example of saliency score distribution.

## A LLM USAGE STATEMENT

We clarify that this paper does not use LLMs for research ideation or paper writing.

## B DETAILED ALGORITHMS

---

**Algorithm 2:** MergeSort

---

**Input:** sorted frame groups $SF_{(k-1)m+1}^{km}$, $k \in [1, \lceil \frac{D}{m} \rceil]$, video summary $S_D$
**Output:** sorted frames $SF_1^D$

1 mid=$\lceil \frac{D}{2m} \rceil$, Sorted=[] ;

2 $A = SF_1^{\lceil \frac{D}{2} \rceil}$=MergeSort(1,mid) ;                    ▷ Binary recursion

3 $B = SF_{\lceil \frac{D}{2} \rceil+1}^{D}$=MergeSort(mid+1,$\lceil \frac{D}{m} \rceil$) ;

4 **while** *A and B are not exhausted* **do**

5    $C_1^{\frac{m}{2}}$, $C_{k\frac{m}{2}}^{k_m}$=LLM($A_1^{\frac{m}{2}}$, $B_1^{\frac{m}{2}}$, $S_D$) ;                    ▷ Equ. 3

6    **if** $len(A + B) \le m$ **then** Sorted.append($C_1^m$) ;

7    **else** Sorted.append($C_1^{\frac{m}{2}}$) ;

8    Put $C_{k\frac{m}{2}}^{k_m}$ back to $A$ and $B$;

9 **end**

10 Sorted.append(Remaining $SF$);

11 **return** Sorted

---

The detailed ranking process is in Algorithm 2. We leverage the typical binary recursion to iterate the frames. To merge two sorted groups, we first select $m/2$ frames from each group, and then we query the LLM to update the total $m$ frames. Following typical merge-sort, we only save the first half of the sorted $m$ frames to enable subsequent LLM comparison.

## C CASE STUDY OF SALIENCY SCORE

We present an example of the estimated score in a soccer game in Fig. 10. We can find that all the baselines fail to fit the actual distribution. The plain curve indicates that saliency models only overfit the training videos without truly learning from semantic content, while video understanding models also overlook the most appealing parts. On the contrary, HiVid generally understands the video content with excellent video summary and frame analysis, e.g., low rating (30) during interview and high score (85) during shooting.

Table 9: Time overhead for ABRs w/ different module delays when end-to-end latency $T = 5s$.

| Baselines | Raw ABR | HiVid | w/ LLM | w/ prediction | w/ both |
|---|---|---|---|---|---|
| ABR Time/s $\downarrow$ | 0.486 | 0.494 | 10.316 | 1.836 | 11.616 |
| Proportion of $T$ $\downarrow$ | 9.72% | 9.88% | 206.32% | 36.72% | 232.32% |

Table 11: Ablation study of HiVid w/ different settings and parameters. Blue and Red denote the best and worst for each metric across the HiVid variants. The default standard deviation $\sigma = 5$.

| Metric / Method | Youtube-8M | | | | TVSum | | | | SumMe | | | |
|---|---|---|---|---|---|---|---|---|---|---|---|---|
| | PLCC | SRCC | mAP50 | mAP15 | PLCC | SRCC | mAP50 | mAP15 | PLCC | SRCC | mAP50 | mAP15 |
| HiVid | 0.66 | 0.67 | 0.86 | 0.53 | 0.50 | 0.52 | 0.67 | 0.40 | 0.47 | 0.47 | 0.62 | 0.37 |
| HiVid w/ last frame | 0.66 | 0.68 | 0.86 | 0.52 | 0.51 | 0.51 | 0.66 | 0.40 | 0.47 | 0.48 | 0.62 | 0.37 |
| HiVid w/ middle frame | 0.65 | 0.66 | 0.84 | 0.50 | 0.51 | 0.52 | 0.68 | 0.41 | 0.47 | 0.47 | 0.61 | 0.36 |
| HiVid w/ $2\sigma$ | 0.68 | 0.69 | 0.89 | 0.55 | 0.52 | 0.55 | 0.70 | 0.41 | 0.49 | 0.49 | 0.64 | 0.38 |
| HiVid w/ $\frac{\sigma}{2}$ | 0.62 | 0.62 | 0.81 | 0.50 | 0.47 | 0.48 | 0.65 | 0.38 | 0.45 | 0.44 | 0.59 | 0.35 |

## D  END-TO-END LATENCY

To demonstrate our adaptive prediction for real time requirement, we present the time overhead in Table 9. We can find that HiVid only imposes additional 8ms which stems from asynchronous rating and prediction. Therefore HiVid achieves the same latency as the original ABR. However, without our adaptive prediction, we would have to await the significant delay to prepare all the historical input for forecasting, which leads to much higher latency (11.616s). For an example of latency $T = 5s$, the decision time of HiVid+ABR does not impact overall experience, while the delay waiting would completely block the ABR decision.

## E  MORE ABLATION STUDY: DIFFERENT ABRS

Fig. 7 in our main paper has shown that HiVid (w/ MPC) surpasses various saliency model baselines with higher PLCC in mean opinion score (MOS) correlation. We also apply HiVid on RL-based Pensieve Mao et al. (2017), IL-based Comyco Huang et al. (2019) and compare with traditional QoE-free buffer-based (BB) and rate-based (RB) ABRs. For RL and IL-based, we incorporate the future $N$ chunk weights as input and train the model to capture the content-aware preference in QoE. We also modify the reward into weighted QoE

Table 10: MOS correlation $\uparrow$ w/ different ABR algorithms.

| ABR | | PLCC$\uparrow$ | SRCC$\uparrow$ |
|---|---|---|---|
| w/ QoE model | w/o QoE model | | |
| HiVid+Pensieve | | 0.78 | 0.79 |
| Pensieve | / | 0.76 | 0.76 |
| HiVid+Comyco | | 0.85 | 0.85 |
| Comyco | | 0.81 | 0.81 |
| / | BB | 0.68 | 0.69 |
| | RB | 0.73 | 0.72 |

to guide the ABR exploration. The MOS correlation for different ABRs is in Table 10.

Note that RB and BB do not incorporate a QoE model and thus cannot be combined with our QoE weights from HiVid. We can find that ABRs with HiVid enhancement outperform those without our method. HiVid with Comyco performs the best because the ABR itself achieves the SOTA traditional QoE optimization Huang et al. (2019). BB is the worst because it only applies buffer to monitor and predict future evolution based on a heuristic parameter.

## F  MORE ABLATION STUDY: DIFFERENT PARAMETERS

We also conduct experiments for different anchor frame choices and Gaussian smoothing parameters in Table 11. The results demonstrate that frame sampling method only introduces limited performance deviation, because a video chunk of 1 second often comprises many semantic-similar frames, regardless of the specific position. While the Gaussian smoothing can present significant impact. Overall, higher deviation $\sigma$ means smoother curve and more stable ratings and hence better performance. However, a significantly high $\sigma$ can also eliminate the original information of our

Table 12: Accuracy consistency across runs.

| Metric | HiVid w/ GPT-4o | HiVid w/ Gemini-2-flash | HiVid w/ Grok2 | HiVid w/ Claude-3-haiku |
|---|---|---|---|---|
| PLCC↑ | 0.66±0.03 | 0.60±0.03 | 0.62±0.02 | 0.53±0.05 |
| SRCC↑ | 0.67±0.03 | 0.59±0.04 | 0.61±0.02 | 0.55±0.04 |
| mAP50↑ | 0.86±0.02 | 0.81±0.05 | 0.82±0.02 | 0.81±0.03 |
| mAP15↑ | 0.53±0.01 | 0.50±0.02 | 0.51±0.01 | 0.48±0.01 |

ranking module and thus render a plain curve. We choose $\sigma = 5$ for more stable performance without losing the fine-grained details of our ranking.

## G  MORE ABLATION STUDY: RANKING CONSISTENCY

We have demonstrated that HiVid can be combined with local mllm in Table 4, which ensures consistency across runs by setting $do\_sample = False$ and seeds. We evaluate the robustness of HiVid with proprietary LLMs across 5 runs in Table 12. The low standard

Table 13: Accuracy consistency across models.

| Metric | GPT-4o & Gemini2 | Gemini2 & Grok2 | GPT-4o & Grok2 |
|---|---|---|---|
| PLCC | 0.87±0.06 | 0.91±0.04 | 0.89±0.04 |
| SRCC | 0.89±0.05 | 0.90±0.04 | 0.90±0.02 |

variance proves the stable correlation and detection accuracy within each model. This means that LLMs generate most of the chunk-level scores via confident zero-shot subjective reasoning rather than random guess, e.g. the justified rating for each frame in Fig. 10.

While Table 13 further evaluates the accuracy between models, where Gemini and Grok exhibit similar results with PLCC=0.91. However, this does not necessarily imply better performance on video saliency score, i.e. GPT-4o outperforms other LLMs in Table 5.

## H  MORE ABLATION STUDY: DETAILED PERFORMANCE FOR LIVE STREAMING

We have demonstrated the clean forecasting performance in Table 3. We also present the prediction with actual LLM rating against the second SOTA Crossformer in Table 14. It shows that inaccurate LLM output (as forecasting input) degrades the overall accuracy compared with clean GroundTruth input, especially for correlation PLCC. This is expected since we

Table 14: Forecasting ablation with LLM output.

| Metric | Forecast-only | | Forecast w/ LLM output | |
|---|---|---|---|---|
| | HiVid-M | Crossformer | HiVid-M | Crossformer |
| MAE↓ | 0.08 | 0.09 | 0.13 | 0.16 |
| RMSE↓ | 0.12 | 0.12 | 0.18 | 0.22 |
| PLCC↑ | 0.29 | 0.23 | 0.20 | 0.15 |
| SRCC↑ | 0.27 | 0.22 | 0.19 | 0.13 |

trained the models with clean data, because using LLM output requires significant offline rating generation.

We also present the forecasting accuracy with module ablation in Table 15. We find that the our content attention effectively learns the interdependent relationships from the various modalities (PLCC accuracy gains of 0.03), while the only uni-modal model performs the worst without specific guidance from video content.

Table 15: Forecasting module ablation.

| Metric | HiVid-M | HiVid-M w/o Attention | HiVid-U |
|---|---|---|---|
| MAE↓ | 0.08 | 0.08 | 0.08 |
| RMSE↓ | 0.12 | 0.12 | 0.12 |
| PLCC↑ | 0.29 | 0.26 | 0.24 |
| SRCC↑ | 0.27 | 0.25 | 0.22 |

## I  MORE ABLATION STUDY: MOS CONSISTENCY AND STATISTICAL SIGNIFICANCE

Table 17: Statistical Significance of MOS correlation from 30 participants.

| | Metric | GT | HiVid | DETR | VideoLLaMA3 |
|---|---|---|---|---|---|
| PLCC | Value ↑ | 0.88 | 0.76 | 0.61 | 0.63 |
| | p value ↓ | $10^{-38}$ | $10^{-25}$ | $10^{-14}$ | $10^{-18}$ |
| | CI | [0.83,0.92] | [0.67,0.82] | [0.48,0.73] | [0.51,0.71] |
| SRCC | Value ↑ | 0.91 | 0.77 | 0.65 | 0.65 |
| | p value ↓ | $10^{-46}$ | $10^{-31}$ | $10^{-18}$ | $10^{-20}$ |

To demonstrate the robustness of user study, we recruit more volunteers and conduct inter-rater consistency check in Table 16. Coefficient of Variation (CV) equals to (std/mean)*100, and we derive CV for each video and then compute the averaged CV. We also leverage Intraclass

Table 16: Inter-rater consistency of 30 participants.

| Metric | Coefficient of Variation↓ | ICC(2,k)↑ | ICC(A,k)↑ |
|---|---|---|---|
| Result | 19.76% | 0.82 | 0.73 |

Correlation Coefficient (ICC) to analyze the robustness. ICC(2,k) evaluates the relative rating consistency while ICC(A,k) is more strict that includes the inter-rater variance. We can find that $< 20\%$ CV and $0.82 > 0.75$ are considered "good" according to Koo & Li (2016). While ICC(A,k) is also "moderately" good, demonstrating valid user study.

In addition, we also present the PLCC and significance for MOS correlation in Table 17. We can find that the results are similar to that in Fig. 7, and our p values for both PLCC and SRCC across different baselines are near zero, demonstrating convincing user rating and our model QoE.

## J    PROMPT INSTRUCTIONS

We present the prompts used during perception module for basic periodical summary and group ratings.

---

Prompt:

I have uploaded {len(image_path)} frames, each representing a video chunk of 1 second. You first extract the frame number attached below the image content. These frames exhibit a continuous {len(image_path)} seconds video clip. The original video background for title and category are {info}. Before this video clip, the periodical video summary is: {story_last}.

Your task is as follows:

1. Based on the frames, periodical summary and background, summarize what story this video has conveyed so far and output your answer as "story_total". (No more than 100 words)

2. Based on the summary and frames, on a scale of integer (0,100), rate all the {len(image_path)} frames such that higher score exhibits higher interestingness score. Different frames can yield the same scores.

Your answer must be a json format like this:

json

[

("story_partial": "xxx"),

("story_total": "xxx"),

json

[

("frame": xxx, "rating": xxx),

("frame": xxx, "rating": xxx),

---

```
("frame": xxx, "rating": xxx)
]
]
```

## K    DISCUSSIONS

We present some clarification for HiVid regarding design and evaluations.

**1. The usage of LLMs simulates human ratings regarding QoE model.**

As explained in Related Work in Section 2.2, LLMs have been widely leveraged for subjective tasks. For example, Fei et al. (2024) propose Vitron to perform image editing based on prompt understanding and image analysis. It has also been demonstrated in Hussain et al. (2024) how LLMs correlate to human behavior regarding subjective perception, including the overall understanding of real world information like images and texts.

Therefore, our proposal to leverage LLMs for subjective video rating also makes sense regarding human-centric QoE modeling

**2. We leverage existing SOTA LLMs instead of training from scratch.**

As explained in Section 2.2, proprietary SOTA LLMs such as GPT-4o Openai (2025) and Gemini Google (2025) have dominated the text generation domain. Training open-source models like Llama Meta (2025) also suffices but would require tremendous time and computation overhead. More importantly, we focus on real world application where ABRs are deployed at the client side. These devices possess limited CPU/GPU resources and cannot afford local LLMs inference.

Nonetheless, it remains a promising direction to fine-tune a new LLM specifically on various datasets like Youtube-8M Sul et al. (2023) in future work.

## L    LIMITATION AND FUTURE WORK

Despite the SOTA results of our HiVid on multiple datasets, HiVid comprises 2 limitations: (1) The overall inference time overhead for VOD is slightly longer than sliding window iteration, because the ranking module adopts binary recursion to enable sufficient comparisons among different frames to ensure accuracy. One future direction is deriving better LLM-guided sorting algorithm that achieves $O(n)$ time complexity at the cost of more storage like Bucket Sort.

(2) We only test HiVid in the video streaming scenario. However, the preference weights in a video can also benefit other vision-language-action applications where human judgment also plays a vital role. Therefore, another future direction is applying our core idea of LLM-guided rating and ranking in video compression. For example, we can derive frame-level saliency score to apply different R-D-$\lambda$ parameters. In this way, we reallocate the bitrates in regions that are appealing to humans.

