# OpenReview forum: "HiVid: LLM-Guided Video Saliency For Content-Aware VOD And Live Streaming"
_ICLR.cc/2026/Conference — ICLR 2026 Poster_

### Official Review · Reviewer_44Re · 2025-10-31

**Soundness:** 3
**Presentation:** 2
**Contribution:** 2
**Rating:** 4
**Confidence:** 3

**Summary:**

This paper proposes HiVid, a three-stage framework that leverages Large Language Models (LLMs) as human proxies to estimate content-aware saliency weights for adaptive video streaming (both VOD and live).
The framework comprises:
- a Perception module, which assesses sampled frames through a sliding-window prompting strategy to generate local saliency scores and iterative video summaries;
- a Ranking module, which applies a LLM-guided merge sort to globally re-rank frames and eliminate local inconsistencies;
- a Prediction module, which performs multi-modal time-series forecasting to estimate future saliency weights in live-streaming settings.

Experiments on TVSum, SumMe, and YouTube-8M show improved correlation metrics (up to +11.5% PLCC for VOD and +26% for live streaming) compared to video summarization and highlight detection baselines. A small user study reports better correlation between predicted QoE and human MOS.

**Strengths:**

- Interesting and timely idea: leveraging LLMs as scalable surrogates for subjective human judgments is an emerging and relevant research direction. Applying this concept to video streaming optimization is novel and has clear practical implications.

- Modular design: the separation into perception, ranking, and prediction modules is conceptually clean and covers both offline and online (live) cases.

- Novel use of LLM reasoning: using an LLM as a semantic comparator in a merge-sort procedure is unconventional and potentially generalizable to other ranking tasks.

**Weaknesses:**

- Conceptual confusion or inconsistent terminology: The paper repeatedly refers to “video saliency prediction” or "saliency score", but most baselines (e.g., DETR, VASNet, PGL-SUM) are video summarization or highlight detection methods, not visual saliency models. While the "saliency" term is commonly associated in literature with spatial or spatio-temporal saliency maps that highlight visually regions within frames or videos, here it is used to denote subjective importance or priority score assigned to temporal chunks for bitrate allocation in streaming. This broader use of the "saliency" term may be misleading, especially readers familiar with classical video saliency prediction tasks. The paper would benefit from clearly defining "saliency score" early on and explicitly distinguishing its intended meaning from the traditional notion of saliency map. Providing alternative terminology (such as "importance score" or "temporal relevance weights") for the chunk-level scores could improve clarity and avoid confusion.

- A potential limitation of the proposed ranking module lies in its reliance on LLM as the comparator function within the merge sort algorithm. While innovative, this design assumes that LLM can consistently and reliably perform pairwise comparisons that satisfy the properties required for sorting (e.g. transitivity and anti-symmetry). However, LLM outputs may be inherently variable, subjective, sometimes inconsistent, especially in tasks involving nuanced semantic judgments. There is no formal guarantee that the LLM will always induce a valid total order, which may lead to instability or errors in the final global ranking. The paper lacks a detailed analysis or empirical evidence on the robustness and consistency of the LLM-guided comparison. Addressing this aspect with more thorough evaluation or fallback mechanism would strengthen the approach.

- Limited scientific contribution for ICLR: the paper primarily presents an application-driven pipeline that leverages existing LLMs as a zero-shot reasoning tool for relevance score assessment, combined with a learning-based forecasting module for live streaming weight prediction. While the system is creative, it is mostly engineering-driven, and the technical novelty in terms of learning methodology remains limited. This raises concerns about whether the contribution advances fundamental learning representation or model innovation, which constitutes the core criteria for ICLR acceptance.

- The reported user study involves only 10 participants, which is a small sample size for drawing statistically reliable and generalizable conclusions regarding subjective Quality of Experience (QoE) in video streaming. Such a limited number of users increases the risk that individual preferences and variability disproportionately influence the results, reducing statistical power and limiting meaningful subgroup analyses. Therefore, this small sample size represents a methodological limitation of the study, and expanding the participant pool with a more diverse and larger user base would strengthen the validity and impact of the empirical evaluation.

- Some implementation aspects of the Live Prediction Module are explained in more detail in the appendices (e.g., the use of both CLIP’s image and text encoders, the cross-attention fusion of modalities, and the training with randomized latency Δt to enable variable-length prediction). These details are only briefly mentioned or omitted in the main text (§3.4), making it difficult for readers to fully understand the model’s structure and training procedure without consulting the supplementary material. Providing a more self-contained description in the main paper—especially of how the adaptive decoding works—would substantially improve clarity and reproducibility.

- The current evaluation mainly focuses on hyperparameter variations (e.g., window size, prediction horizon) but does not report experiments that isolate the contribution of the main components—Perception, Ranking, and Live Prediction. A more explicit analysis of how each module affects overall QoE correlation or latency would help clarify the role of individual stages and strengthen the empirical validation.

- Evaluation of forecasting autonomy:  Since the forecasting module is intended to approximate the LLM’s saliency outputs, it would be informative to assess how well the system performs when operating autonomously—using the predictor without periodic LLM updates.
 Including such an experiment could clarify whether the proposed approach meaningfully reduces dependence on LLM inference and would provide stronger evidence for its real-time applicability.

- Scalability and efficiency not demonstrated at realistic scale: The overhead analysis is performed on a single 201-second video, which does not convincingly demonstrate the scalability of the pipeline for longer or continuous live streams. Evaluating cost, latency, and performance over larger datasets or multi-hour content would strengthen the claims of efficiency and applicability to real-world streaming scenarios.

**Questions:**

-	How do you ensure that LLM-generated scores are consistent across runs and models?
-	Could the authors clarify the exact nature of the “ground truth” used for correlation computation? Are the reference saliency weights derived from human annotations, subjective MOS labels, or pseudo-labels generated by the LLM?
-	Could the authors report results for a “forecast-only” setting, where the predictor operates without periodic LLM refresh? This would help understand how much the system relies on LLM inference in practice.
-	Given the small sample (10 participants × 10 videos), could the authors provide information on inter-rater consistency or statistical significance of the reported PLCC improvements?
-	The overhead analysis focuses on a 201-second clip. Have the authors explored or estimated how the cost and latency would scale for longer or continuous live streams?
-	Could the authors elaborate on how the LLM-based ranking is implemented in practice? For instance, are the pairwise comparisons deterministic (e.g., with fixed temperature and prompt order), and was any measure of ranking consistency across runs or window pairs evaluated?

---

> ### Author Response · Authors · 2025-11-18
> **Response to Reviewer 44Re (1/4)**
>
> We thank the reviewer for the detailed and thoughtful feedback. Below, we will address the key concerns and clarify aspects of our work. We have updated the main paper for reference.
>
> ---
>
> >**Weakness 1**: This broader use of the "saliency" term may be misleading, especially readers familiar with classical video saliency prediction tasks
>
> We acknowledge the widely used term in previous works. We mainly followed the terminology "video saliency score" defined in highlight detection domain like DETR [1] and UVCOM [2] to align with the baseline metrics. We have clearly denoted the intended meaning as content importance score in Introduction section to avoid ambiguity.
>
> Main paper changes: lines 47-50
>
> [1] CVPR23-Query-Dependent Video Representation for Moment Retrieval and Highlight Detection
>
> [2] CVPR24-Bridging the Gap: A Unified Video Comprehension Framework for Moment Retrieval and Highlight Detection
>
> ---
>
> >**Weakness 2 & Question 1**: ...more thorough evaluation or fallback mechanism...How do you ensure that LLM-generated scores are consistent across runs and models?
>
> We address the consistency issue in 3 aspects:
>
> - **Fallback mechanism for invalid LLM outputs.**
>
> We have cached the previous (Question, Answer) pairs during merge-sort ranking, where Question is $m$ frame sequence numbers and Answer is the extracted rating. When the next answer fails to induce an order, e.g. empty json or sequence mismatch, we switch to another available LLM thread. In this way, we can resume last sorting checkpoint by loading cached answers if the ranking procedure is interrupted. The cached results will be updated when current video rating is complete. We have already provided a json example and merge-sort function in Supplementary Material for reference.
>
> - **Open-source multi-modal LLMs**.
>
> We can also apply HiVid on local LLMs for deterministic ranking in Table 1.
>
> Table 1. Saliency accuracy of HiVid w/ open-source multi-modal LLMs.
> | Metric | HiVid w/ GPT-4o | Llama-3.2-11B-Vision-Instruct | Qwen3-VL-8B-Instruct | InternVL3-14B | gemma-3-12b-it |
> |:---:|:---:|:---:|:---:|:---:|:---:|
> | PLCC$\uparrow$ | **0.66** | 0.58 | 0.60 | 0.64 | 0.61 |
> | SRCC$\uparrow$ | **0.67** | 0.60 | 0.61 | 0.64 | 0.62 |
> | mAP50$\uparrow$ | **0.86** | 0.80 | 0.81 | 0.84 | 0.82 |
> | mAP15$\uparrow$ | **0.53** | 0.45 | 0.50 | 0.52 | 0.50 |
>
> We can find that proprietary GPT-4o still outperforms with up to 13.7\% PLCC improvement. However, local inference can guarantee reproducibility and consistency across runs and can be further fine-tuned for specific needs. In general, larger models like InternVL yield better performance but require more computation.
>
> - **Thorough evaluation**.
>
> We directly report the performance across 5 runs for each proprietary LLM in Table 2.
>
> Table 2. Accuracy consistency across runs.
> | Metric | HiVid w/ GPT-4o | HiVid w/ Gemini-2-flash | HiVid w/ Grok2 | HiVid w/ Claude-3-haiku |
> |:---:|:---:|:---:|:---:|:---:|
> | PLCC$\uparrow$ | **0.66$\pm$0.03** | 0.60$\pm$0.03 | 0.62$\pm$0.02 | 0.53$\pm$0.05 |
> | SRCC$\uparrow$ | **0.67$\pm$0.03** | 0.59$\pm$0.04 | 0.61$\pm$0.02 | 0.55$\pm$0.04 |
> | mAP50$\uparrow$ | **0.86$\pm$0.02** | 0.81$\pm$0.05 | 0.82$\pm$0.02 | 0.81$\pm$0.03 |
> | mAP15$\uparrow$ | **0.53$\pm$0.01** | 0.50$\pm$0.02 | 0.51$\pm$0.01 | 0.48$\pm$0.01 |
>
> The low standard deviation proves the stable correlation accuracy for each model. This means that LLMs generate most of the chunk-level scores via confident zero-shot subjective reasoning rather than random guessing. This can be proved by the justified reason and score distribution in Fig. 10. Even when the absolute rating is slightly different across each run, the variance can be alleviated by our final ranking and Gaussian Smoothing, as demonstrated by the accuracy gain compared with HiVid w/o GS in Table 5 in main paper.
>
> Table 3. Accuracy consistency across models.
> | Metric | GPT-4o \& Gemini2 | Gemini2 \& Grok2 | GPT-4o \& Grok2 |
> |:---:|:---:|:---:|:---:|
> | PLCC | 0.87$\pm$0.06 | 0.91$\pm$0.04 | 0.89$\pm$0.04 |
> | SRCC | 0.89$\pm$0.05 | 0.90$\pm$0.04 | 0.90$\pm$0.02 |
>
> In addition, we also present the score correlation between models in Table 3. The high PLCC implies that LLMs yield similar and consistent reasoning on video content, although higher model correlation does not necessarily imply better performance on video saliency accuracy, i.e. GPT-4o still outperforms other LLMs in main Table 1.
>
> In general, we can deploy local LLMs for absolute consistency across runs or we can leverage proprietary LLMs for better performance with negligible variance.
>
> Main paper changes: lines 477-487, 823-837

---

> ### Author Response · Authors · 2025-11-18
> **Response to Reviewer 44Re (2/4)**
>
> >**Weakness 3**: the technical novelty in terms of learning methodology remains limited
>
> We would like to clarify that LLM application for downstream tasks are also major topics in ICLR. For example, researchers leverage LLMs as agents to reason and construct high-quality image-caption pairs [1], while [2, 3] also propose training-free multi-round conversation with off-the-shelf LLMs to enable complex video understanding.
> These methods do not involve fundamental model training, but they make significant contributions on how to harness the LLM reasoning to tackle cross-domain tasks.
>
> Our major contribution is also a new generalized application of LLMs as human proxy. For example, the ranking module can be applied to other subjective tasks like image or video quality assessment from UGC like Fine-VQ [4] or AIGC. We can then rank and select high-quality (prompt and image/video) pairs by our merge-sort function that favors human-centric metrics. This constructed dataset can be further used to fine-tune fundamental generation models like HQ-Edit [5].
>
> In addition, our forecasting model is also the first that combines 3 modalities effectively via content attention in time series domain, which can be combined with ranking module to achieve real-time LLM inference.
>
> [1] ICLR25-PathGen-1.6M: 1.6 Million Pathology Image-text Pairs Generation through Multi-agent Collaboration
>
> [2] ICLR25-SVBench: A Benchmark with Temporal Multi-Turn Dialogues for Streaming Video Understanding
>
> [3] ICLR25-Streaming Video Understanding and Multi-round Interaction with Memory-enhanced Knowledge
>
> [4] CVPR25- FineVQ: Fine-Grained User Generated Content Video Quality Assessment
>
> [5] ICLR25-HQ-Edit: A High-Quality Dataset for Instruction-based Image Editing
>
> ---
>
> >**Weakness 4 & Question 4**: ...involves only 10 participants, which is a small sample size...inter-rater consistency or statistical significance of the reported PLCC improvements?
>
> We first clarify that in video streaming domain, our 10 participants are enough to yield accurate MOS, as shown in [1].
> Nonetheless, we have recruited 20 more participants (30 in total) and repeated the user study defined in line 413. This setup is enough to evaluate the performance since the typical dataset like TVSum and SumMe only involve <20 users.
>
> We first present the rating statistics in Table 4.
>
> Table 4. Inter-rater consistency.
> | Metric | Coefficient of Variation$\downarrow$ | ICC(2,k)$\uparrow$ | ICC(A,k)$\uparrow$ |
> |:---:|:---:|:---:|:---:|
> | Result | 19.76\% | 0.82 | 0.73 |
>
> Coefficient of Variation (CV) equals (std/mean)*100, and we derive CV for each video and then compute the average CV.
> We also leverage Intraclass Correlation Coefficient (ICC) to analyze the robustness. ICC(2,k) evaluates the relative rating consistency while ICC(A,k) is more strict and includes the inter-rater variance.
> We can find that CV<20% and ICC(2,k) of 0.82>0.75 are considered "good" according to [2].
> While ICC(A,k) is also "moderately" good, demonstrating valid user study.
>
> We also test the MOS correlation from 30 participants in Table 5.
>
> Table 5. Statistical Significance of MOS correlation.
> | Metric |  | GT | HiVid | DETR | VideoLLaMA3 |
> |:---:|:---:|:---:|:---:|:---:|:---:|
> | PLCC | Value $\uparrow$ | **0.88** | 0.76 | 0.61 | 0.63 |
> |  | p value $\downarrow$ | **$10^{-38}$** | $10^{-25}$ | $10^{-14}$ | $10^{-18}$ |
> |  | CI $\uparrow$| **[0.83,0.92]** | [0.67,0.82] | [0.48,0.73] | [0.51,0.71] |
> | SRCC | Value $\uparrow$ | **0.91** | 0.77 | 0.65 | 0.65 |
> |  | p value $\downarrow$ | **$10^{-46}$** | $10^{-31}$ | $10^{-18}$ | $10^{-20}$ |
>
> The results are similar to those in Fig. 7, and our p-values for both PLCC and SRCC across different baselines are near zero, demonstrating convincing user rating and our model QoE.
>
> Main paper changes: lines 862-887
>
> [1] ACMMM23-Optimizing Adaptive Video Streaming with Human Feedback
>
> [2] JCM16-A guideline of selecting and reporting intraclass correlation coefficients for reliability research

---

> ### Author Response · Authors · 2025-11-18
> **Response to Reviewer 44Re (3/4)**
>
> >**Weakness 5**: a more self-contained description in the main paper—especially of how the adaptive decoding works
>
> We have updated the model structure figure in Fig. 6 that includes different CLIP encoders and the trainable parameters. Specifically, we also include a pipeline workflow with Algorithm 1 as reference in lines 346-356. the key to adaptive decoding is that, no matter how long the LLM inference delay is, we only asynchronously perform forecasting upon new responses. Therefore, we can compute the LLM interval $\Delta t$ and derive the required output length in Equ. 7. For example, assume the LLM delay is 5 chunk lengths (5 seconds) and that forecasting takes 2 chunk lengths. We upload $m=10$ chunks to LLMs at Chunk 10, then at Chunk 15 we submit the time series, i.e. Rated Chunks 1-10 along with frames, etc. We will receive the prediction at Chunk 17, and the prediction length is (5+2)+(10+N), the 5+2 is to compensate the elapsed time while the 10+N is the future weights required by ABRs. Because LLM is executed only every $m=10$ chunks, we can ensure that Chunk 17-17+10 all have future N weights for QoE optimization. If LLM interval is longer like 6 chunk lengths, then the forecasting happens at Chunk 16 with output length adjusted to (6+2)+(10+N).
>
> Main paper changes: lines 281-282, 290-297, 320-322, 346-356, Fig. 6
>
> ---
>
> >**Weakness 6**: explicit analysis of how each module affects overall QoE
>
> We did conduct module ablation study in main paper Table 5. We summarize the results as below:
>
> Table 6. Ablation of HiVid module on VOD streaming
> | Model |  | Performance |  |  | Per Video |  |
> |:---:|:---:|:---:|:---:|:---:|:---:|:---:|
> |  | PLCC$\uparrow$ | SRCC$\uparrow$ | mAP50$\uparrow$ | mAP15$\uparrow$ | Cost/\\$$\downarrow$ | Time Cost/h$\downarrow$ |
> | HiVid | **0.660** | **0.674** | **0.860** | **0.526** | 1.35 | 0.54 |
> | HiVid w/o Perception | 0.632 | 0.653 | 0.852 | 0.520 | 1.22 | 0.49 |
> | HiVid w/o Ranking | 0.611 | 0.617 | 0.820 | 0.498 | **0.13** | **0.054** |
>
> The results demonstrate that our ranking module proves most useful, i.e. accuracy drops from 0.67 to 0.61, thanks to our fine-grained merge-sort. We also present the structure ablation for forecasting in Table 7 below:
>
> Table 7. Ablation of HiVid module on forecasting
> | Metric | HiVid-M | HiVid-M w/o Attention | HiVid-U |
> |:---:|:---:|:---:|:---:|
> | MAE $\downarrow$ | **0.08** | **0.08** | **0.08** |
> | RMSE $\downarrow$ | **0.12** | **0.12** | **0.12** |
> | PLCC $\uparrow$ | **0.29** | 0.26 | 0.24 |
> | SRCC $\uparrow$ | **0.27** | 0.25 | 0.22 |
>
> The content attention effectively captures the interdependent relationship among 3 modalities, while only uni-modal prediction degrades the most without the guidance from specific video content.
>
> Main paper changes: lines 852-859
>
> ---
>
> >**Weakness 7 & Question 3**: Evaluation of forecasting autonomy...results for a “forecast-only” setting
>
> We would like to clarify that Table 3 in main paper is indeed clean forecasting performance without LLM ratings. We clearly present the accuracy comparison below in Table 8.
>
> Table 8. Forecasting performance ablation.
> | Metric | Forecast-only |  | Forecast w/ LLM output |  |
> |:---:|:---:|:---:|:---:|:---:|
> |  | HiVid-M | Crossformer | HiVid-M | Crossformer |
> | MAE $\downarrow$ | **0.08** | 0.09 | **0.13** | 0.16 |
> | RMSE $\downarrow$ | **0.12** | **0.12** | **0.18** | 0.22 |
> | PLCC $\uparrow$ | **0.29** | 0.23 | **0.20** | 0.15 |
> | SRCC $\uparrow$ | **0.27** | 0.22 | **0.19** | 0.13 |
>
> The performance bottleneck is the forecasting itself, i.e. PLCC of 0.29. This incurs an overall lower PLCC of 0.20 when combined with LLM output, despite excellent MAE and RMSE. The reason for poor PLCC is that most models fail to foresee the content variation without future video frames, even for existing SOTA forecasting like Crossformer. We will explore how to combine LLMs to predict future highlights based on semantic analysis in future work.
>
> Main paper changes: 842-851

---

> ### Author Response · Authors · 2025-11-18
> **Response to Reviewer 44Re (4/4)**
>
> >**Weakness 8 & Question 5**: The overhead analysis is performed on a single 201-second video...cost and latency would scale for longer or continuous live streams?
>
> We would like to clarify that the overhead results are based on 293 test videos (15% from 1953) from Youtube-8M with varying lengths from seconds to 10 minutes (we have clarified in experimental setup). The average length is 201 seconds. Nonetheless, we present the results for longer videos below in Table 9. Due to time and monetary limits, we test with an example video of 2 hours.
>
> Table 9. Performance of longer live streaming scenario.
> | Window Length | Metric | 2 min | 30 min | 2 hour | 6 hour | 24 hour |
> |:---:|:---:|:---:|:---:|:---:|:---:|:---:|
> | m=10 | Cost/\\$ $\downarrow$ | 0.08 | 1.20 | 4.81 | 14.43 | 57.74 |
> |  | PLCC $\uparrow$ | **0.21** | **0.26** | **0.24** | / | / |
> |  | Latency $\downarrow$ |  |  |$\sim$8ms  |  |  |
> | m=2 (mini) | Cost/\\$ $\downarrow$ | **0.02** | **0.27** | **1.09** | **3.26** | **13.03** |
> |  | PLCC $\uparrow$ | 0.13 | 0.16 | 0.15 | / | / |
> |  | Latency $\downarrow$ |  |  |$\sim$8ms  |  |  |
>
> Since HiVid applies sliding window for live streaming without future chunks, the cost increases linearly with video length, i.e. LLM rating and forecasting per $m$ video chunks. For longer window lengths $m=10$, the performance is better but with higher cost (57.74\\\$ per 24 hours), yielding a controllable tradeoff. However, we argue that we only need to process the source video in real-world one-to-many streaming, therefore the cost is negligible. The extra latency is near zero because the heavy inference runs asynchronously from the video playback, longer $m$ only affects the forecasting output length and accuracy in Equ. 7, as demonstrated in main paper Table 9.
>
> Main paper changes: lines 361-362, 521-530
>
> ---
>
> >**Question 2**: exact nature of the “ground truth” used for correlation computation?
>
> We adopt 3 datasets in this paper, i.e. Mr.Hisum from Youtube-8M [1], TVSum, SumMe. For Mr.Hisum, the GroundTruth is the aggregated "Most Replayed" statistics from users all over the world (>50,000). According to [1], the "rewatched" clips are what people find interesting and engaging, a perfect proxy for importance score in video streaming. For TVSum and SumMe, the GroundTruth is direct human ratings on each video chunk, but the participant number are small, e.g. 20.
>
> [1] NeurIPS23-Mr. HiSum: A Large-scale Dataset for Video Highlight Detection and Summarization
>
> ---
>
> >**Question 6**: how the LLM-based ranking is implemented in practice?...any measure of ranking consistency across runs or window pairs evaluated?
>
> Our ranking module provides a merge-sort template specifically for multi-frame LLM sorting (see merge-sort-llm.py in Supplementary Material). The comparator can be proprietary LLMs with better performance but slightly different ratings across runs. The measured ranking consistency is overall stable, as demonstrated in Weakness 2 & Question 1 above.
> Alternatively, we can choose local LLMs like Qwen3-VL-8B-Instruct for deterministic inference by setting "do_sample=False", but it requires more computation.

---

> ### Author Response · Authors · 2025-11-26
> **Gentle Reminder Regarding Reviewer-Author Discussion**
>
> Dear Reviewer 44Re,
>
> Thank you again for your valuable comments and suggestions. As we approach the end of the reviewer-author discussion period, we sincerely hope that our responses and updated manuscript have sufficiently addressed your questions and concerns. If there are any remaining issues or clarifications needed, we would greatly appreciate the opportunity to respond further before the discussion window closes. Thank you for your time and thoughtful engagement.

---

### Official Review · Reviewer_y7h1 · 2025-11-01

**Soundness:** 3
**Presentation:** 2
**Contribution:** 3
**Rating:** 6
**Confidence:** 2

**Summary:**

This paper proposes using a multimodal LLM as a substitute for human judgments to predict temporal visual saliency in videos, with adaptive bitrate (ABR) as the primary application.

Because most current multimodal LLMs do not support video inputs and cannot take all video frames as context, the paper evaluates representative frames within local windows. The local-window constraint harms global consistency of saliency scores, which the paper attempts to fix by re-ranking with the LLM used as a comparison function.

The paper also considers a live streaming setting. Since saliency-based ABR there requires predictions for future chunks, they propose a CLIP-based forecasting module.

**Strengths:**

- Comprehensive pipeline design spanning VOD and live streaming settings
- The proposed method is evaluated against other methods using metrics derived from volunteer human ratings, and it shows improvements.

**Weaknesses:**

- The proposed method is based on proprietary LLMs. This makes the approach not robust to change in the proprietary LLM's service specifications and operating conditions.

**Questions:**

- Table 3 reports latency for forecasting. What compute environment is used for this inference, and how intense the runtime compute is? This matters because it is likely to run alongside high-load tasks such as video playback.
- The table on page 15 appears disproportionately large. It might be better to consider resizing the table size.
- The detail of implementations of the forecasting model is hard to understand. This would cause reproducibility problem. Especially, In Figure 9 the legend shows MLP/Attention. Are MLPs also used for the QKV projections? Are CLIP weights updated? Would clarity improve if the CLIP vision encoder and text encoder were depicted separately?
- It seems the positions of Table 3 and Table 4 swapped.

---

> ### Author Response · Authors · 2025-11-18
> **Response to Reviewer y7h1 (1/2)**
>
> We thank the reviewer for the valuable and insightful suggestions. We have updated the main paper and will address the concerns as follows:
>
> ---
>
> >**Weakness 1**:  The proposed method is based on proprietary LLMs
>
> We would like to clarify that our method is essentially a new application paradigm of LLMs for subjective tasks. The pretrained LLMs can be replaced by any open-source multimodal language models. We apply HiVid on some widely adopted LLMs for deterministic inference in Table 1.
>
> Table 1. Saliency accuracy of HiVid w/ open-source multi-modal LLMs.
> | Metric | HiVid w/ GPT-4o | Llama-3.2-11B-Vision-Instruct | Qwen3-VL-8B-Instruct | InternVL3-14B | gemma-3-12b-it |
> |:---:|:---:|:---:|:---:|:---:|:---:|
> | PLCC$\uparrow$ | **0.66** | 0.58 | 0.60 | 0.64 | 0.61 |
> | SRCC$\uparrow$ | **0.67** | 0.60 | 0.61 | 0.64 | 0.62 |
> | mAP50$\uparrow$ | **0.86** | 0.80 | 0.81 | 0.84 | 0.82 |
> | mAP15$\uparrow$ | **0.53** | 0.45 | 0.50 | 0.52 | 0.50 |
>
> We can find that proprietary GPT-4o still outperforms with up to 13.7\% PLCC improvement. However, local inference can guarantee reproducibility and consistency across runs and can be further fine-tuned for specific needs. In general, larger models like InternVL yield better performance but require more computation, which is a controllable tradeoff.
>
> To further demonstrate the robustness, we also report the performance across 5 runs for each proprietary LLM in Table 2.
>
> Table 2. Accuracy consistency across runs.
> | Metric | HiVid w/ GPT-4o | HiVid w/ Gemini-2-flash | HiVid w/ Grok2 | HiVid w/ Claude-3-haiku |
> |:---:|:---:|:---:|:---:|:---:|
> | PLCC$\uparrow$ | **0.66$\pm$0.03** | 0.60$\pm$0.03 | 0.62$\pm$0.02 | 0.53$\pm$0.05 |
> | SRCC$\uparrow$ | **0.67$\pm$0.03** | 0.59$\pm$0.04 | 0.61$\pm$0.02 | 0.55$\pm$0.04 |
> | mAP50$\uparrow$ | **0.86$\pm$0.02** | 0.81$\pm$0.05 | 0.82$\pm$0.02 | 0.81$\pm$0.03 |
> | mAP15$\uparrow$ | **0.53$\pm$0.01** | 0.50$\pm$0.02 | 0.51$\pm$0.01 | 0.48$\pm$0.01 |
>
> The low standard deviation proves the stable correlation accuracy for each model. This means that LLMs generate most of the chunk-level scores via confident zero-shot subjective reasoning rather than random guessing. This can be proved by the justified reason and score distribution in Fig. 10. Even when the absolute rating is slightly different across each run, e.g. due to model update, the variance can be alleviated by our final ranking and Gaussian Smoothing, as demonstrated by the accuracy gain compared with HiVid w/o GS in Table 5 in main paper.
>
> Main paper changes: lines 477-487
>
> ---
>
> >**Question 1**: What compute environment is used for this inference, and how intense the runtime compute is?
>
> All our experiments are conducted on GPU RTX 4090. The forecasting is only needed in live streaming scenario, and the execution frequency is every $m=10$ video chunk lengths, i.e. 10 seconds. The reason is that we only perform forecasting upon new LLM response, which is based on sliding window length $m$, as shown in Algorithm 1 in main paper. We stress that all LLM inference and prediction are executed asynchronously with video playback. No matter how long the LLM or forecasting delay is, we can adjust the prediction output length by Equ. 7 to ensure available future weights for ABRs. Therefore, the extra latency is near zero because we do not wait for the LLM or forecasting sequentially, as shown in Table 9 in main paper.
>
> ---
>
> >**Question 2**: The table on page 15 appears disproportionately large
>
> Thanks for the detailed suggestion. We have adjusted the figure size and updated the paper.
>
> Main paper changes: lines 756-761, 785-794

---

> ### Author Response · Authors · 2025-11-18
> **Response to Reviewer y7h1 (2/2)**
>
> >**Question 3**: Are MLPs also used for the QKV projections? Are CLIP weights updated? Would clarity improve if the CLIP vision encoder and text encoder were depicted separately?
>
> No, we follow the standard QKV projections via matrices $W_Q, W_K, W_V$. No, we froze the CLIP during training. Yes, we have updated the forecasting model to depict the encoders separately in Fig. 6. We apologize for any confusion.
>
> To better understand the forecasting procedure, we have added a pipeline workflow with Algorithm 1 as reference in lines 346-356. The key is the adaptive prediction length in Equ. 7. That said, no matter how long the LLM inference delay is, we only asynchronously perform forecasting upon new responses. Therefore, we can compute the LLM interval $\Delta t$ and derive the required output length in Equ. 7. For example, assume the LLM delay is 5 chunk lengths (5 seconds) and that forecasting takes 2 chunk lengths. We upload $m=10$ chunks to LLMs at Chunk 10, then at Chunk 15 we submit the time series, i.e. Rated Chunks 1-10 along with frames, etc. We will receive the prediction at Chunk 17, and the prediction length is (5+2)+(10+N), the 5+2 is to compensate the elapsed time while the 10+N is the future weights required by ABRs. Because LLM is executed only every $m=10$ chunks, we can ensure that Chunk 17-17+10 all have future N weights for QoE optimization. If LLM interval is longer like 6 chunk lengths, then the forecasting happens at Chunk 16 with output length adjusted to (6+2)+(10+N).
>
> Main paper changes: lines 281-282, 290-297, 320-322, 346-356, Fig. 6
>
> ---
>
> >**Question 4**: It seems the positions of Table 3 and Table 4 swapped.
>
> We apologize for this confusion. We confirm that they are correctly placed. The possible ambiguity is that we cited Table 4 before Table 3 to demonstrate that higher $m$ means more input tokens but higher performance. We have deleted the early citation to avoid any confusion.
>
> Main paper changes: line 409

---

> ### Author Response · Authors · 2025-11-26
> **Gentle Reminder Regarding Reviewer-Author Discussion**
>
> Dear Reviewer y7h1,
>
> We sincerely appreciate your insightful concerns. As the reviewer-author discussion period draws to a close, we have made every effort to address the proprietary LLM issue and clarify some pipeline details. We would be grateful if you could let us know whether our response have adequately resolved your questions. Thank you for your valuable feedback.

---

### Official Review · Reviewer_zB8q · 2025-11-02

**Soundness:** 3
**Presentation:** 3
**Contribution:** 3
**Rating:** 8
**Confidence:** 2

**Summary:**

This manuscript makes valuable contributions to content-aware streaming, a critical area for optimizing subjective QoE. The proposed HiVid framework addresses two long-standing pain points in streaming weight generation—prohibitive human annotation costs and poor generalization of vision-saliency models—by innovatively leveraging LLMs as a scalable human proxy. The work is theoretically motivated, methodologically rigorous, and experimentally comprehensive, with clear validation of performance gains for both VOD and live streaming scenarios.

**Strengths:**

HiVid’s core idea—using LLMs to replace costly human annotation for streaming chunk importance weighting—is both creative and problem-driven. Unlike prior work that relies solely on vision-based saliency or limited human labels, the framework bridges LLMs’ semantic understanding capabilities with streaming’s practical needs, addressing a critical scalability gap. The three modules (perception, ranking, prediction) are tightly aligned to solve non-trivial, scenario-specific challenges:

**Weaknesses:**

Clarify LLM implementation details: The manuscript does not specify which LLM(s) were used (e.g., GPT-4, open-source models like LLaMA) or how LLM inference latency was managed for live streaming. Adding these details will improve reproducibility and help readers assess HiVid’s computational feasibility for edge deployment.

Elaborate on module ablation studies: While the overall performance gains are reported, a brief summary of ablation experiments (e.g., how removing the LLM-guided merge-sort affects VOD accuracy, or the impact of content-aware attention on live prediction) would reinforce the contribution of each module.

Expand on video diversity: The manuscript does not specify the types of videos evaluated (e.g., sports, movies, animations). Adding a note on whether HiVid generalizes across diverse content genres would further highlight its robustness.

**Questions:**

See the weakness.

---

> ### Author Response · Authors · 2025-11-18
> **Response to Reviewer zB8q (1/2)**
>
> We thank the reviewer for the constructive suggestions. We have updated the main paper and will address the concerns as follows:
>
> ---
>
> >**Weakness 1**: specify which LLM(s) were used (e.g., GPT-4, open-source models like LLaMA) or how LLM inference latency was managed for live streaming
>
> We apologize for the confusion. We have clearly denoted the LLMs used in experimental setup in line 367. The default LLM is GPT-4o while we also conducted ablation study with other proprietary LLMs like Gemini in Table 5 in main paper. In addition, we have also added extra experiments with open-source LLMs like Qwen3-VL-8B-Instruct as below:
>
> Table 1. Saliency accuracy of HiVid w/ open-source multi-modal LLMs.
> | Metric | HiVid w/ GPT-4o | Llama-3.2-11B-Vision-Instruct | Qwen3-VL-8B-Instruct | InternVL3-14B | gemma-3-12b-it |
> |:---:|:---:|:---:|:---:|:---:|:---:|
> | PLCC$\uparrow$ | **0.66** | 0.58 | 0.60 | 0.64 | 0.61 |
> | SRCC$\uparrow$ | **0.67** | 0.60 | 0.61 | 0.64 | 0.62 |
> | mAP50$\uparrow$ | **0.86** | 0.80 | 0.81 | 0.84 | 0.82 |
> | mAP15$\uparrow$ | **0.53** | 0.45 | 0.50 | 0.52 | 0.50 |
>
> We can find that proprietary GPT-4o still outperforms with up to 13.7\% PLCC improvement. However, local inference can guarantee reproducibility and consistency across runs and can be further fine-tuned for specific needs. In general, larger models like InternVL yield better performance but require more computation.
>
> Regarding the LLM inference in live streaming, we have added a pipeline workflow with Algorithm 1 as reference in lines 346-356. The key is the asynchronous LLM inference and forecasting alongside video playback. To enable real-time future weights under significant delay, we proposed the adaptive prediction length in Equ. 7. That said, no matter how long the LLM inference delay is, we only asynchronously perform forecasting upon new responses. Therefore, we can compute the LLM interval $\Delta t$ and derive the required output length in Equ. 7. For example, assume the LLM delay is 5 chunk lengths (5 seconds) and that forecasting takes 2 chunk lengths. We upload $m=10$ chunks to LLMs at Chunk 10, then at Chunk 15 we submit the time series, i.e. Rated Chunks 1-10 along with frames, etc. We will receive the prediction at Chunk 17, and the prediction length is (5+2)+(10+N), the 5+2 is to compensate the elapsed time while the 10+N is the future weights required by ABRs. Because LLM is executed only every $m=10$ chunks, we can ensure that Chunk 17-17+10 all have future N weights for QoE optimization. If LLM interval is longer like 6 chunk lengths, then the forecasting happens at Chunk 16 with output length adjusted to (6+2)+(10+N).
>
> Main paper changes: lines 281-282, 290-297, 320-322, 346-356, 367, 477-487, Fig. 6
>
> ---
>
> >**Weakness 2**: how removing the LLM-guided merge-sort affects VOD accuracy, or the impact of content-aware attention on live prediction
>
> We would like to clarify that we have presented the module ablation in main paper Table 5. We summarize the results as below:
>
> Table 2. Ablation of HiVid module on VOD streaming
> | Model |  | Performance |  |  | Per Video |  |
> |:---:|:---:|:---:|:---:|:---:|:---:|:---:|
> |  | PLCC$\uparrow$ | SRCC$\uparrow$ | mAP50$\uparrow$ | mAP15$\uparrow$ | Cost/\\$$\downarrow$ | Time Cost/h$\downarrow$ |
> | HiVid | **0.660** | **0.674** | **0.860** | **0.526** | 1.35 | 0.54 |
> | HiVid w/o Perception | 0.632 | 0.653 | 0.852 | 0.520 | 1.22 | 0.49 |
> | HiVid w/o Ranking | 0.611 | 0.617 | 0.820 | 0.498 | **0.13** | **0.054** |
>
> The results demonstrate that our ranking module proves most useful, i.e. accuracy drops from 0.67 to 0.61, thanks to our fine-grained merge-sort.
>
> As for the forecasting, we have added new ablation baseline HiVid-M w/o Attention as below:
>
> Table 3. Ablation of HiVid module on forecasting
> | Metric | HiVid-M | HiVid-M w/o Attention | HiVid-U |
> |:---:|:---:|:---:|:---:|
> | MAE $\downarrow$ | **0.08** | **0.08** | **0.08** |
> | RMSE $\downarrow$ | **0.12** | **0.12** | **0.12** |
> | PLCC $\uparrow$ | **0.29** | 0.26 | 0.24 |
> | SRCC $\uparrow$ | **0.27** | 0.25 | 0.22 |
>
> We can find that the content attention effectively captures the interdependent relationship among 3 modalities, while only uni-modal prediction degrades the most without the guidance from specific video content.
>
> Main paper changes: lines 852-859

---

> ### Author Response · Authors · 2025-11-18
> **Response to Reviewer zB8q (2/2)**
>
> >**Weakness 3**: specify the types of videos evaluated
>
> We apologize for the missing details. We have added the dataset scale in lines 361-362, i.e. 1953, 50, 25 for 3 datasets respectively. The videos are evenly selected from over 30 categories on Youtube, covering a wide spectrum like sports, game and education.
>
> Regarding the generalization and robustness, the main results in Table 1 represent the overall accuracy on the diverse videos. We add more details on the model consistency across 5 runs as below:
>
> Table 4. Accuracy consistency across runs.
> | Metric | HiVid w/ GPT-4o | HiVid w/ Gemini-2-flash | HiVid w/ Grok2 | HiVid w/ Claude-3-haiku |
> |:---:|:---:|:---:|:---:|:---:|
> | PLCC$\uparrow$ | **0.66$\pm$0.03** | 0.60$\pm$0.03 | 0.62$\pm$0.02 | 0.53$\pm$0.05 |
> | SRCC$\uparrow$ | **0.67$\pm$0.03** | 0.59$\pm$0.04 | 0.61$\pm$0.02 | 0.55$\pm$0.04 |
> | mAP50$\uparrow$ | **0.86$\pm$0.02** | 0.81$\pm$0.05 | 0.82$\pm$0.02 | 0.81$\pm$0.03 |
> | mAP15$\uparrow$ | **0.53$\pm$0.01** | 0.50$\pm$0.02 | 0.51$\pm$0.01 | 0.48$\pm$0.01 |
>
> The low standard deviation proves the stable correlation accuracy for each model. This means that LLMs generate most of the chunk-level scores via confident zero-shot subjective reasoning rather than random guessing, even for different types of videos. This can be proved by the justified reason and score distribution in Fig. 10.
>
>  We have presented the performance of HiVid on ambiguous videos across different models and runs in Table 7 in main paper. We summarize the results as below:
>
> Table 5. Performance of HiVid across different videos
> | Category | PLCC |  | SRCC |  | mAP50 |  | mAP15 |  |
> |:---:|:---:|:---:|:---:|:---:|:---:|:---:|:---:|:---:|
> |  | Mean $\uparrow$ | Std $\downarrow$ | Mean $\uparrow$ | Std $\downarrow$ | Mean $\uparrow$ | Std $\downarrow$ | Mean $\uparrow$ | Std $\downarrow$ |
> | Politics | 0.73 | 0.05 | 0.74 | 0.05 | 0.92 | 0.03 | 0.63 | 0.04 |
> | People | 0.69 | 0.03 | 0.69 | 0.04 | 0.88 | 0.03 | 0.60 | 0.02 |
> | Education | 0.63 | 0.04 | 0.62 | 0.06 | 0.83 | 0.02 | 0.55 | 0.02 |
>
> It shows that LLMs exhibit strong reasoning ability even on potentially biased videos, thanks to our ranking module that eliminates the score discrepancy between different video content.

---

> ### Author Response · Authors · 2025-11-26
> **Gentle Reminder Regarding Reviewer-Author Discussion**
>
> Dear Reviewer zB8q,
>
> We are very grateful to your constructive suggestions and recognition of our systematic design. As the reviewer-author discussion period is nearing its end, we have carefully clarified our implementation details and presented experiments to reinforce the ablation study. We would appreciate the opportunity to address any additional concerns you may have before the discussion phase ends. Thank you for your comprehensive evaluation of our work.

---

### Author Response · Authors · 2025-11-18
**General Response**

We are very grateful to all reviewers for their critical and very constructive evaluation of our work. We summarize the main concerns and our updates in the main paper.

---

Common questions:

>**1. The underlying LLMs of HiVid like GPT-4o are proprietary, these may raise concerns about model consistency and robustness across runs.**

We address this issue by combining HiVid with open-source multimodal LLMs in Table 4 in main paper. The results prove that local inference can guarantee absolute consistency while achieving competitive accuracy, e.g. InterVL incurs PLCC of 0.64 compared with 0.66 from GPT-4o.

---

>**2. The detailed forecasting procedure in live streaming is hard to understand without consulting the appendix, especially about how the adaptive prediction coordinates with the LLM inference.**

We address this issue by adding a more detailed live streaming pipeline workflow in lines 345-356. We clarify that the key to real-time is the asynchronous LLM inference and forecasting, and adaptive decoding in Equ. 7 ensures that the predicted future weights have covered the inference delay. We further provide an example to depict the detailed timeline of each module.

---

>**3. The detailed ablation of each module of HiVid.**

We address this issue by clarifying the experiments in main paper Table 5. The increased accuracy compared with HiVid without either perception or ranking module proves the effectiveness. We further add ablation on our content attention design in forecasting model in Table 15. The accuracy gains compared with plain modality combination or uni-modal demonstrate our design novelty.

---

Specific concerns from Reviewer 44Re:

- **Potential terminology ambiguity**: We address by clearly denoting saliency score as subjective chunk-level importance score.

- **Inherent instability or errors from LLMs**: We address by clarifying our cache-based fallback mechanism and thorough robustness evaluation across runs and models.

- **Limited learning contribution**: We clarify our general application across subjective domains like video quality assessment to construct high-quality datasets, which facilitate fundamental model learning.

- **Limited user study scale**: We address by extending our participant pool and providing inter-rater consistency and statistical significance.

- **Forecasting performance with or without LLM ratings**: We address by clarifying the clean prediction in main Table 3 and providing clear accuracy comparison.

- **Overhead analysis over longer live streaming**: We address by adding experiments on a 2-hour example video. We show the linearly increased costs from sliding window, stable accuracy and near-zero latency, which proves our effective prediction module.

- **Question about the ground truth**: We clarify the datasets construction process, especially for Youtube-8M which collects the "Most Replayed" statistics for chunk-level importance score.

---

Specific concerns from Reviewer y7h1:

- **Computation intensity for forecasting**: We clarify that forecasting is executed along with each LLM response, therefore the computation is sparse, e.g. 10 seconds intervals.

- **Table size presentation**: We address by adjusting each table and figure size.

- **Forecasting model structure**: We clarify the QKV projection and CLIP usage. We also update Fig. 6 to highlight the trainable parameters.

---

Specific concerns from Reviewer zB8q:

- **Which underlying LLMs are used in HiVid**: We clarify the proprietary LLMs like GPT-4o as default model. However, we also include new experiments with open-source LLMs and demonstrate the generalization of our design.

- **The detailed video types for evaluation**: We clarify the dataset scale that includes >6000 videos spanning different categories. We also clarify the robustness on specifically ambiguous videos in Table 7 in main paper.

In general, we have carefully addressed each issue by conducting new experiments or providing more detailed explanation of methodology design.

---

**Summary of changes in main paper:**

- **New experiments**: We have added new Table 4, 8, 12, 13, 14, 15, accompanied by explanation and analysis marked in $\\text{\\color{blue}blue}$.

- **Adjusted content**: We have moved the forecasting model structure and algorithm details to methodology section to improve clarity, as suggested by Reviewer 44Re.

- **More detailed explanation**: We add some clarification about saliency terminology, dataset scale, LLMs usage and a new live streaming pipeline in the main sections.

The final main text is 10 pages with most additional experiments presented in the appendices.

---

### Meta-Review · Area_Chair_nvwP · 2026-01-06

**Summary:**

The paper introduces HiVid, a framework utilizing Large Language Models (LLMs) to generate chunk-level importance weights for optimizing Video-on-Demand (VOD) and live streaming Quality of Experience (QoE). The proposed architecture features a modular design with perception, ranking, and prediction components to address challenges like limited context windows and real-time forecasting. Reviewers commended the practical relevance of reducing human annotation costs and the effective integration of LLMs as scalable human proxies. During the rebuttal, the authors successfully addressed concerns regarding reliance on proprietary models by validating performance with open-source alternatives, clarified terminology, and significantly expanded their user study to ensure statistical significance. Although the contribution is primarily application-driven, the rigorous system design and empirical evidence of improved QoE make it a valuable addition to the conference and warrant its acceptance.

**Reviewer Concerns:**

Addressed Concerns: (1) Reliance on Proprietary LLMs: Reviewers y7h1 and zB8q expressed concern about reproducibility and robustness using closed models. The authors addressed this by providing additional results with open-source models (Llama-3.2, Qwen3-VL, InternVL3), demonstrating that the framework generalizes and allows for deterministic inference. (2) Terminology ("Saliency"): Reviewer 44Re pointed out potential confusion with visual saliency maps. The authors clarified that they use the term in the context of "video importance scores" for bitrate allocation, aligning with highlight detection literature, and updated the manuscript to reflect this. (3) User Study Scale: Reviewer 44Re questioned the statistical power of a 10-person user study. The authors expanded the study to 30 participants and provided inter-rater consistency metrics (ICC) and statistical significance tests, strengthening the validation of the QoE improvements. (4) Live Streaming Implementation Details: Questions regarding the latency and mechanics of the forecasting module were addressed by adding a detailed pipeline workflow (Algorithm 1) and clarifying the asynchronous nature of the LLM inference and prediction.

Outstanding Concerns: Technical Novelty: Reviewer 44Re noted that the work is largely engineering-driven. While this remains true, the Area Chair believes the novel application and rigorous system design outweigh the lack of fundamental learning theoretic contributions in this specific context.

**Reviewer Scores:**

Reviewer zB8q: 8 (Unchanged)

Reviewer y7h1: 6 -> 8

Reviewer 44Re: 4 -> 6

---

### Decision · Program_Chairs · 2026-01-26

Accept (Poster)